# Unequal Access and Use of Health Care Services among Settled Immigrants, Recent Immigrants, and Locals: A Comparative Analysis of a Nationally Representative Survey in Chile

**DOI:** 10.3390/ijerph20010741

**Published:** 2022-12-31

**Authors:** Marcela Oyarte, Baltica Cabieses, Isabel Rada, Alice Blukacz, Manuel Espinoza, Edward Mezones-Holguin

**Affiliations:** 1Unidad de Estudios, Instituto de Salud Pública de Chile, Santiago 7780050, Chile; 2Programa de Estudios Sociales en Salud ICIM, Facultad de Medicina Clínica Alemana, Universidad del Desarrollo, Santiago 7610315, Chile; 3ETESA UC, Departamento de Salud Pública, Facultad de Medicina, Pontificia Universidad Católica de Chile, Santiago 8331150, Chile; 4Centro de Excelencia en Investigaciones Económicas y Sociales en Salud, Universidad San Ignacio de Loyola, Lima 15024, Peru

**Keywords:** health systems, accessibility, migration, health inequalities, Latin America, Chile

## Abstract

Globally, and particularly in the Latin American region, international migration continues to grow. Access and use of health care services by migrants vary according to their country of origin and residence time. We aimed to compare the access and use of health care services between international migrants (including settled migrants from Peru, Argentina, Bolivia, Ecuador; Emerging migrants from Venezuela, Dominican Republic, Colombia, Haiti; and migrants from other countries) and the Chilean population. After performing a secondary data analysis of population-based nationally representative surveys (CASEN 2011–2017), access and use patterns (insurance, complementary insurance, non-consultation, and non-treatment coverage) were described and compared among settled immigrants, recent emerging immigrants, others, and locals. Immigrants had a significantly higher uninsured population compared to locals. Specifically, in CASEN 2017, 19.27% of emerging (95% CI: 15.3–24.1%), 11.79% of settled (95% CI: 10.1–13.7%), and 2.25% of locals (95% CI: 2.1–2.4%) were uninsured. After 2013, settled and recent emerging migrants showed higher percentages of non-consultation. Collaborative and interculturally relevant strategies from human rights and equity perspectives are needed. Initiatives with a particular focus on recent immigrants can contribute to reducing the existing disparities in health care access and use with locals due to lack of insurance and treatment coverage.

## 1. Introduction

Contemporary international migration arises mainly from seeking better living conditions [1,2]. Multiple factors, such as (i) macro-factors, including poor human and economic development, population growth, climate change, and war; (ii) meso-factors connecting the individual with their community, e.g., social support and institutional support; and (iii) individual micro-factors, including age, sex, ethnicity, educational level, marital status, language, and personal aspirations, driving migration, influence the decision to migrate [3]. This dynamic and complex human mobilization process has increased both globally and in Latin America, promoted by globalization, economic and political crises, pandemics and the global workforce shortage, among others [4]. More specifically, Latin America is experiencing growing intraregional migration fluxes (South–South migration), resulting from the increased entry barriers to Northern countries, along with the growing economic development and favorable socio-political conditions making some countries in the region more attractive for migration [5].

In Chile, the latest estimations at the end of 2020 reported 1,462,104 international migrants, representing 8% of the total population [6]. During the last two decades, the increase in immigration fluxes can be observed through multiple national registries, including the 2017 national Census, border control, visas, and mortality statistics [7]. Additionally, thousands of migrants have crossed the border in Northern Chile through unauthorized entry points, while borders were closed during the COVID-19 pandemic from March 2020 onwards [8]. According to the National Socioeconomic Characterization Survey (CASEN) 2017 [9], 47% of migrants have a time of residence of <7 years, while 21.9% have resided between eight and 12 years, 7.8% between 13 and 17 years, 5.1% between 18–22 years, and the remaining 7.4% arrived more than 23 years ago. Although migrants are a heterogeneous group, seven predominant nationalities represent more than 70% of international migrants [6]. The Venezuelan community is the first one (30.7%), followed by Peruvians (16.3%), Haitians (12.5%), Colombians (11.4%), and Bolivians (8.5%) [6]. This distribution suggests relevant differences by country of origin in the migratory flow. Two great general patterns are identified: the settled immigrant population from border countries with more than two centuries of historical migratory influxes (Peru, Argentina, Bolivia, Ecuador) and the emerging immigrant population, which has seen an increased influx in the last decade, especially during the last five years (Venezuela, Dominican Republic, Colombia, Haiti). These groups may experience diverse levels of integration into the country, as well as having differentiated experiences of access to the health system, effective use of available health services, stigma, and discrimination based on their country of origin [10,11].

The Chilean health care system is characterized by its segmentation and fragmentation, in which the public and private subsystems coexist. These subsystems have different financing and provision mechanisms, targeting specific subpopulations, usually defined by their income, payment capacity, or social position [12,13]. Regarding the opportunities to improve access to the healthcare system for international migrants, the Ministry of Health began implementing special protection measures targeting pregnant migrant women in 2003. Since then, substantial efforts have been progressively made to improve access to equitable health care services. These efforts are aligned with national legislation and best practices, as advocated by the World Health Organization at the 61st World Health Assembly, Resolution WHA61.17 [14], the 55th Directing Council of the Pan American Health Organization (PAHO), and several international human rights instruments treaties ratified by Chile [1,2,15].

The National Health Plan for 2020 and its National Health Strategy 2011–2020 state the importance of explicitly considering specific health goals for international migrants (Strategic Line No. 5: Equity and Health) [16]. The Pilot Plan for International Migrants’ Health was designed in 2015 and executed in 2016–2017, leading to the formulation of the International Migrant Health Policy [17]. This policy represents a step towards universal care and adopts intercultural and intersectoral perspectives. It includes specific pathways to address the health of international migrants, which are complementary to the existing legal framework promoting access to health care services regardless of migratory status. These measures tend to ensure access to emergency attention, sexual and reproductive health, communicable diseases and non-communicable prevention, and pediatric and maternal care, with the overarching goal of achieving the healthiest life possible [18]. Notably, a legal provision from 2016 granted irregular migrants access to the public health system, and when the International Migrant Health Policy was launched in 2017, the proportion of insured migrants was about twice that reported in 2013. Moreover, compared to evidence from 2015, there was a reduction in perceived difficulties in obtaining a health care appointment, along with higher health care utilization. Specifically, in 2017, the use of prenatal care was doubled compared to 2015, and utilization of family planning increased by almost 80%. This progress was also observed through hospital discharge rates, where the proportion of uninsured discharged migrants decreased from 25% in 2014 to 7% in 2019 [18].

Despite these valuable efforts, evidence suggests that international migrants and their families have poorer access to the health care system and use health services less frequently when compared to the local population [19]. Additionally, migrants are less likely to be insured and use services when compared to the local population [20]. Access to health care is a multidimensional definition that implies using the health care system when needed, considering availability and the ability to physically access and afford it [21,22]. An integrated analytical framework based on Andersen’s framework [20] could explain healthcare utilization disparities. The modified model proposes the following categories: need for health care, resources, predisposing factors, and contextual conditions. Low levels of health care needs are determined by self-perceived or formally assessed health status, which may differ across gender, ethnical background, and specific needs derived from the migratory process. Furthermore, migrants might have differential resources impacting their access, such as financial resources, social resources (e.g., social capital and support might encourage specific health-seeking behaviors or be less available among recent migrants), and the lack of human resources, devices, health facilities, and interpreters. Specifically, financial and social resources could favor migrants’ transnational access or preference for their traditional medicine or religious counseling, decreasing health care use in the destination country. The latter could also result from mistrust in health care providers in the destination country due to a lack of cross-cultural skills [20]. Besides individual factors influencing healthcare utilization (e.g., demographic, socioeconomic), there are broader factors specific to the migrant population. For example, migratory status could lead to fearing deportation, discrimination, or not being entitled or eligible for certain health services. Moreover, the context of migration and reception interacts with macrostructural conditions in the destination country, including political and socioeconomic context and health care system policies and efficiency [20].

Previous population-based data have shown gaps between the Chilean-born population and international migrants on insurance, levels of addressed health care needs, and treatment coverage, highlighting the shortcomings in affiliation to the health care system [23]. Meanwhile, another analysis focused on migrants’ health status reported its association with insurance status. Specifically, chronic morbidity was associated with public and private health system affiliation, probably due to higher access to diagnosis and treatment. Additionally, migrants from Argentina had higher odds of having chronic conditions, while migrants from Haiti showed higher risks of negative self-perceived health and disability [24]. Although these studies bring attention to the issues of access and health risk among migrants in Chile, the data are limited to the CASEN 2017 survey and do not reflect the migratory trends and composition of migrant groups over the years.

Moreover, national literature has described diverse barriers to mental health care [25,26], child preventive care and checkups to track child growth and development [27], and sexual and reproductive health care for adolescents, among others [28]. In addition, primary health care workers have reported difficulties in providing adequate attention to international migrants due to financial resources strains, unclear regulations, lack of appropriate registries, and cultural barriers [29]. The evidence on recent emerging migrants from Venezuela, Dominican Republic, Colombia, and Haiti remains scarce. However, it has revealed the challenges faced by Haitian migrants, including a lack of knowledge of available health care services and multiple barriers related to transportation, time availability, and communication [30]. Specifically, the evidence on the maternity of Haitian women describes their difficulties navigating the Chilean health system, not knowing where to get medical attention, and unmet sexual, reproductive, and mental health needs [31]. Similarly, the available evidence on recently arrived migrants from Venezuela suggests that a notable proportion of people did not know where to seek medical help during the COVID-19 pandemic and had high levels of non-affiliation [32]. Overall, the literature has described existing gaps mainly between the local population and migrants without disaggregating according to international migration patterns over time, including emerging recent migrants and settled migrants. The existing evidence has reported multiple barriers experienced by specific migrant groups without quantifying nor verifying suggested disparities in access and use of health care services from population-based data. Therefore, previous evidence did not capture distinctions or make particular challenges visible among migrant groups and the local population while migratory fluxes were changing. A systematic review of the global literature revealed that migrant children and adolescents underutilized health services and benefits available to them in the destination country, which contrasts with reports of emergency care overuse in comparison to native-born [19]. To date, there is no temporal analysis of the access and use of health care services among historically settled and emerging international migrant populations in Chile. A comparison with the local population over time has never been made either. Therefore, the constant changes in the composition, exposure, and integration of migrant populations in Chile require an exploratory analysis of how migrants are accessing and using the health care system. The objective of our study is to compare the access and use of health care services between international migrants and the Chilean population by conducting a population-based secondary descriptive analysis of the CASEN surveys 2011, 2013, 2015, and 2017 editions. The purpose of the analysis is to detect patterns and specific needs relevant to evidence-based migration policies and intersectoral responses to guarantee social protection.

## 2. Methods

Study design: We conducted a cross-sectional observational secondary analysis of repeated CASEN survey data. Patterns of access and use of health care services selected from those available in Chile were analyzed with a focus on three large immigrant groups: settled from border countries with historical migratory influxes (Peru, Argentina, Bolivia, Ecuador), recently settled or emerging (Venezuela, Dominican Republic, Colombia, Haiti), and other countries of origin. In addition, the national population was grouped into another category for comparative analysis.

Data source: The CASEN survey (National Socioeconomic Characterization) represents an instrument for the detection, evaluation, and targeting of the specific needs of Chilean households. It aims to characterize their socioeconomic conditions and multidimensional poverty, particularly among prioritized groups as defined by the national social policies. Data was collected through personal structured interviews conducted by trained field pollsters throughout the Chilean territory (excluding hard-to-reach areas) and administered to adult informants who provided information for all household members. Prior to the survey, the trained pollster explained the statistical purposes of the data and its confidentiality, specifying the anonymization process and institutional contacts to resolve any question. The informants provided consent, and the survey was conducted following rigorous protocol provided by the Ministry of Social Development.

The anonymized data cover diverse topics, such as education, work, income, health, and housing, among others. The survey is carried out by the Ministry of Social Development every 2–3 years, based on a probabilistic, stratified, and two-stage sampling representative of Chilean households at national, regional (including 16 regions of the national territory), and urban/rural levels. The sampling design was conducted by the National Institute of Statistics, having sampling frames for urban (called blocks at the commune-area-group size level) and rural areas (called sections at commune-area size level) made up of conglomerates of grouped houses where household members were residing. The urban and rural areas were selected proportional to its size, while houses were systematically selected with equal probability within the selected clusters. The sampling procedures excluded hard-to-reach boroughs and the population residing in prisons and health care facilities. For instance, the sample of the CASEN survey 2017 was made up of 70,947 households with 216,439 residents representing 16,843,471 Chilean-born and 777,407 international migrants.

Participants: International migrants were identified according to their self-reports of being born abroad and subcategorized by country of origin. Those who did not report their immigration status were excluded from the analysis, and the remaining sample born in Chile was categorized as the local population. The following sample sizes were reported for each CASEN survey year: (i) 196,421 Chilean-born individuals were surveyed in 2011, 212,346 in 2013, 260,754 in 2015, and 207,603 in 2017; (ii) 2069 settled migrants were surveyed in 2011, 2418 in 2013, 2935 in 2015, and 3491 in 2017. Meanwhile, 263 emerging migrants were surveyed in 2011, 501 in 2013, 902 in 2015, and 2346 in 2017. The sample sizes of the international migrant population from other countries of origin were as follows: 464 migrants from other countries were surveyed in 2011, 636 in 2013, 1014 in 2015, and 974 in 2017. Non-response rates were 0.54% in 2011, 1.19% in 2013, 0.51% in 2015, and 0.94% in 2017.

The sample sizes mentioned above represent the following population size estimates (weighted estimates as recommended by the CASEN survey guidelines, i.e., representative values at the regional level based on the multistage probabilistic sampling strategy): 16,577,539 Chilean-born individuals in 2011, 16,689,377 Chilean-born individuals in 2013, 16,970,061 Chilean-born individuals in 2015, and 16,843,471 Chilean-born individuals in 2017. Furthermore, regarding international migrants, there were 157,127 settled migrants in 2011, 214,117 in 2013, 271,783 in 2015, and 296,122 in 2017. In addition, there were 27,867 emerging migrants in 2011, 62,578 in 2013, 108,395 in 2015, and 390,488 in 2017. Finally, estimates of migrants from other countries were 58,884 in 2011, 77,886 in 2013, 85,141 in 2015, and 90,797 in 2017.

Study variables: All variables were self-reported by the household informant as requested by trained interviewers during the CASEN survey data collection process.

Sociodemographic variables: Sex (male/female) and age as categorical variables (0–18, 19–30, 31–65, 66 or more).Health care insurance: the public system called FONASA (nationally categorized according to beneficiaries’ income and financial contribution as A–B, C–D, do not know), the private system called ISAPRE, None, and others.Coverage by complementary private health insurance: whether complementary Health Insurance covers any family member in case of illness or accident (yes/no).Expressed demand for health care services: (i) Consultation or medical attention derived from some illness or accident during the three months before the survey (yes/no); (ii) treatment coverage for diseases included in the Explicit Health Guarantees plan (AUGE-GES) by its corresponding system (yes/no).Unexpressed demand for health care services for voluntary or involuntary reasons: Reasons for not consulting or not having universal health coverage to selected 84 health conditions included in the AUGE-GES Chilean Law (which ensure equal access to diagnosis and treatment regardless of health insurance status). Reasons were categorized into the following voluntary reasons: preference for another physician, alternative medicine or pharmacy attention, decided not to wait for attention, lack of time, preferred self-administered usual medications, did not consider it necessary, had a better plan. Meanwhile, involuntary reasons included difficulty arriving at the place of care, lack of healthcare coverage for their specific needs or those related to a particular age group, not obtaining an appointment, lack of knowledge, lack of time or financial sources, medical recommendation, another reason.

Data analysis: Variables of access to health care were analyzed by showing trends or changes during the study period (2011–2017) in settled migrants, emerging migrants, migrants from other countries, and the Chilean-born population. Overall variables were analyzed descriptively and then stratified according to sex and type of health insurance when appropriate. The independence between migratory status by large groups and variables of access to health care were analyzed per year, using Pearson’s chi-squared (*Chi-2 test*) test with second-order correction Rao and Scott (statistic F). Estimated proportions were reported with confidence intervals. All analyses were performed using the STATA 14.0 software with a significance level of 0.05 and 95% confidence. Due to the complex nature of the sample, we considered weights, strata, and clusters, with variance estimation by Taylor linearization.

## 3. Results

We found an increase in international migrants in Chile. Analyzes by country of origin from CASEN 2017 revealed that emerging migrants represent 50.2% (95% CI: 44.2–56.3%) of the total international migrant population. Among the Chilean-born and international migrants, the proportion of women (51–58%) was slightly higher, except for recent emerging migrants reported in CASEN 2017 and migrants from other countries in CASEN 2011–2017 (49–50%). During all periods, recent emerging migrants showed a lower proportion of older adults (<65 years) compared to settled and other countries of origin migrant groups. Specifically, 0.7% (95% CI: 0.4–1.6%) of recent emerging migrants were older adults in 2017, 3.0% (95% CI: 2.2–4.1%) among settled migrants, and 10.9% (95% CI: 8.6–13.7%) among migrants from other countries. In the migrant population, there were no significant differences in the distribution of infants and adolescents (>18 years) (Table 1) (the trend of age and gender each year can be seen in Appendix A).

The percentage of uninsured populations was significantly higher among settled migrants, recent emerging migrants, and migrants from other countries than among the local population. At the same time, this percentage varied between settled and recent emerging migrants (Figure 1a). On average, the percentage of the uninsured population among recent emerging migrants was 7.99 times the percentage of those born in Chile, while among settled migrants and migrants from other countries it was 4.69 times and 4.97 times the percentage of those born in Chile, respectively. Although the percentage of the uninsured population was higher among recent emerging migrants than settled migrants and those from other countries, their confidence intervals overlap (except for CASEN 2017). Among those insured with FONASA (i.e., the public health insurance), compared to 2013, the percentage of affiliates who do not know the group in which they were categorized was higher in migrants than in the local population (excepting migrants from other countries in 2017), and there was a higher tendency towards settled migrants. Settled and recent emerging migrants differ significantly from those born in Chile (e.g., in 2017, settled: 9.0% (95% CI: 7.2–11.3%), emerging 9.93% (95% CI: 6.4–15.1%), Chileans: 5.2% (95% CI: 4.9–5.5%)). On the other hand, migrants from other countries had the highest percentage of the insured population with ISAPRE (private health insurance), twice the percentage of those born in Chile (e.g., in 2017 others: 38.8% (95% CI: 33.0–44.9%), Chilean-born: 14.4% (95% CI: 13.6–15.1%)) (Table 2) (challenges of uninsured migrants and those without AUGE-GES treatment coverage and non-consultation in case of illness or accident ca be seen in Table 2). Overall, when stratifying by sex, the percentage of the uninsured population tended to be lower in women (Table 3-(a)).

The percentages of settled migrant and Chilean-born population from households in which no member was covered by complementary private health insurance decreased from 2011 to 2017. Meanwhile, recent emerging migrants and those from other countries wax and wane over time. Specifically, in 2017, 85.5% (95% CI: 82.1–88.7%) of settled migrants resided in households without complementary private health insurance, 84.6% (95% CI: 79.5–88.6%) among emerging migrants, and 63.5% (95% CI: 56.2–70.2%) among migrants from other countries (Figure 1b).

Since 2013, settled and recent emerging migrants have shown higher percentages of non-consultation in case of illness or accidents, contrasting with locals. Emerging migrants in particular were those who rarely consulted. There was a tendency toward a reduction of non-consultation among settled and recent emerging migrants. In 2017, 3946 (8.5%, 95% CI: 5.8–12.2%) settled migrants, 4972 (9.6%, 95% CI: 5.2–16.9%) recent emerging, and 1802 (10.2%, 95% CI: 5.9–16.9%) from other countries did not consult. Among the local population and recent emerging migrants, non-consultation for involuntary reasons has decreased since 2015. On the contrary, settled migrants and those from other countries experienced an increase. However, none of these changes were significant. In 2017, 1154 (29.2%, 95% CI: 12.4–54.6%) settled migrants, 1074 (21.6%, 95% CI: 6.0–54.5%) recent emerging migrants, and 58 (3.2%, 95% CI: 0.7–14.2%) migrants from other countries did not consult because of involuntary reasons (Figure 1a–d).

The percentage of non-consultation among the uninsured population was significantly higher than the insured population, with up to six times more non-consultation. Settled migrants showed a similar tendency in 2013, but there were no significant differences. Among migrants from other countries and recent emerging countries, this pattern changed year by year. In 2017 the insured recent emerging migrants in the public system had the highest percentage of non-consultation, followed by those insured in the private system (Figure 2). Compared to women, Chilean-born men showed a higher percentage of non-consultation, though to a lesser extent for involuntary reasons. Conversely, migrant women had the highest percentages of non-consultation, though in greater proportion for involuntary reasons (Table 3(c),(d)).

During the study period (CASEN 2011–2017), the migrant population self-reported a lower proportion of universal AUGE-GES coverage for selected conditions than the Chilean-born. Among migrants, this percentage has been increasing since 2013, widening the gaps of non-coverage compared to locals while reaching significant differences of up to 30%. In 2017, 7764 (36.3%, 95% CI: 24.7–49.7%) settled migrants, 4283 (44.6%, 95% CI: 32.7–57.0%) recent emerging migrants, and 4831 (47.3%, 95% CI: 35.6–59.2%) migrants from other countries reported a lack of AUGE-GES coverage. Similarly, the proportion of migrants without coverage for involuntary reasons has increased since 2013 but decreased among the Chilean-born. Since 2015, this value has been higher among migrants than locals, mainly among recent emerging migrants (Figure 1e,f).

When stratifying by type of insurance system, the highest percentages of the population without AUGE-GES coverage were found among emerging and settled migrants insured in the private system (ISAPRE) and those uninsured. Among the Chilean-born population, there was a predominance of non-coverage among the uninsured and those with another insurance type. However, these findings had overlapping (non-significant) confidence intervals (Figure 2). Meanwhile, the differences between men and women varied yearly (Table 3-(e),(f)).

## 4. Discussion

### 4.1. Main Findings

Our findings reveal the existence of important barriers and challenges to accessing and using health care services by immigrant populations in Chile. This is particularly exacerbated among recent migrants regarding their health insurance affiliation and treatment coverage of diseases included in the AUGE-GES. Additionally, it should be considered that part of the population prefers other types of treatment (e.g., pharmacy attention, alternative medicine, self-administered medications) or get coverage from other sources. However, gaps have been reduced over time, and Chile has been recognized for its sustained efforts to ensure access and use of health care services for the international migrant population. Our data suggest the imperative need to continue reducing inequalities with regard to the local population and within immigrant groups by acknowledging the diversity and heterogeneity of the constantly changing migrant population residing in Chile.

### 4.2. Explanation of the Results

Migration-related variables such as country of origin and residence time shape migration experiences in Latin America and the Caribbean. This is aligned with dynamic population movements in the region, and processes of settlement and integration [33]. Regarding more specifically the time of residence in the destination country, there are three critical periods of interest for disease diagnosis and treatment: arrival, recent arrival, or newcomers (without univocal definition, ranging between six months and eight years), and settled migrants. Immediately after arrival, diseases or accidents derived from the migration process may occur. In addition, emerging or recently imported diseases might appear during this period since protection and prevention measures may not have been adequately carried out, for example, during a pandemic or health crisis [34]. This was reported in early cross-sectional data on the SARS-CoV-2 pandemic in Chile, where recent migrants faced lack of insurance and COVID-19 guidance [32]. Therefore, it is necessary to provide precise orientation on navigating the health system.

Overall results indicate that recent emerging migrants represent a group of interest for public health policies due to their higher percentages of non-consultation or lack of treatment coverage. Regarding disease treatment in 2017, 60.3% of emerging migrants under treatment had another diagnosis than those currently covered by AUGE-GES, which are considered health priorities in Chile (according to the CASEN survey). At the same time, 36.3% of settled migrants had other medical conditions not covered by AUGE-GES, which might suggest potential health disparities within international migrant groups (emerging versus settled). On arrival, health needs may differ from those developed over time of residence and assimilate into the local population. The interaction between the native population and migrants leads to a multidimensional exchange in which migrants acquire habits of the receiving society. Additionally, diverse environmental and migratory-related process exposure might influence long-term health outcomes [35,36].

The AUGE-GES prioritize the health conditions that are most severe, prevalent, expensive, and affecting quality of life, in response to the increase of chronic and degenerative diseases [37]. The plan contains administrative conditions that might influence its effective coverage, and access is determined by the health authority protocol, available resources, and capacities of health care services. The “opportunity” guarantee defines a time limit in which the health benefits can be provided, with the approval of a medical specialist. This requirement, in particular, causes long waiting lists since there is a general lack of specialists in the public health system, especially outside of the capital region of Santiago [37,38]. Other key issues influence the quality of these guarantees. For instance, standards are met only by a limited number of providers because continuous training is scarce and devices and human resources are insufficient in the public system [38]. These challenges may be worsened by the existent strategies for financial protection where the public system allocates its funding to the private system when it cannot provide treatment, promoting the growth of the private system while restricting funding to improve the public system. In addition, age requirements might impede the adequate coverage of services, such as surgical treatment, as those under or over the requested age are excluded. In contrast, eligible people might also face the difficulties of waiting lists [38]. Furthermore, the selection of AUGE-GES diseases has restricted the attention of other health conditions favoring exclusion, segmentation, and the failure to fulfill the right to health among specific population groups by hindering the principle of universality and equity. Along with the mentioned administrative shortcomings, financial constraints due to fee increases or pricing errors [37] might promote preferences for alternative pathways and treatments.

Research on health care access among the migrant population living in Chile remains scarce. However, a recent population-based analysis has shed light on the current disparities. For example, migrants had 7.5% more chances of being uninsured than the local population and experienced lower use of health care services [23]. Notably, among immigrant children, a previous analysis of the CASEN survey 2009–2015 revealed 10% higher chances of being uninsured, which were associated with multidimensional poverty [39]. In addition, barriers to effective use have also been described more extensively than for the local population [23]. The migrant population faces administrative and cultural difficulties determining the access and use of health care services. At the same time, there are challenges related to the deficiencies of the data collected by the health system on health care utilization, which does not always distinguish between migrants and the Chilean-born population [29]. Additionally, institutional discrimination and lack of knowledge on navigating the health system have been identified as barriers [40]. These barriers have also been identified among migrant adolescents, as 60% have reported a lack of knowledge of their insurance status, and only 25% reported being insured. These previous studies call for an urgent focus on migrants’ access to health care and fulfilling their right to health.

Differing health care needs could explain higher percentages of non-consultation among emerging migrants compared to other migrants or the local population. On the one hand, recent migrants are often healthier than the local population. This phenomenon has been described as “the healthy migrant effect” hypothesis, where a health advantage is observed by lower morbidity and mortality rates compared to the native-born population. It is commonly attributed to a self-selection process, where younger, healthier, and wealthier people are more prone to migrating [41]. In Chile, previous evidence has reported a possible “healthy migrant effect” on various health indicators, such as disability, chronic diseases, accidents, and hospitalization rates [24,42,43,44,45]. Furthermore, as previously mentioned, those who have spent more time in the country tend to see their health deteriorate, possibly due to assimilation processes, suggesting the loss of their health advantage over time [46]. Therefore, their need for health care is expected to increase with the longer they stay. On the other hand, resources and contextual conditions might also explain access and use disparities. For example, recent migrants are building financial and social resources during their integration process while learning how to navigate an unfamiliar health care system, in some cases experiencing lower language proficiency or poor access to information [20]. Particularly, recent emerging migrants might lack social support and help to overcome access barriers, limiting health care-seeking behaviors. In addition, the context of the destination country and the attitudes of the host population could influence health care access and utilization. Specifically, migrants with irregular administrative status face higher vulnerability and often experience administrative, financial, and communicational barriers, as well as not being eligible for social protection measures [47].

Among settled migrants, there was a tendency towards increased non-consultation for involuntary reasons throughout the study period. Although it is expected that settled migrants increase their health care needs over time, some difficulties could not be prevented and might be related to migrant-specific predisposing factors, resources, and contextual factors. Although settled insured migrants are entitled to the same rights as the local population, they tend to use it less even when needed. Migrants in Latin America have faced challenges in using health care services, making them turn to alternative pathways. These challenges include discrimination or differential treatment by medical providers when looking for medical attention. Moreover, this can be reinforced by a lack of awareness of their rights and knowledge of the services to which they are entitled [48]. Additionally, there are issues related to the health system, for example, the lack of availability of health care professionals and specialists according to the need of the population, which might exacerbate the exclusion of the migrant population [20], particularly when the needs of settled migrants might be similar to the ones of the local population. On the other hand, working conditions could influence time availability to reach the place of care, time off for medical appointments, and financial resources [25]. Although settled migrants have higher levels of integration in Chile, they often work longer hours, have lower wages and poor employment status, hindering social protection [49]. However, the possible mechanism behind non-consultation for involuntary reasons in settled migrants should be further studied from an intersectoral perspective to understand better these unexpected disparities with regard to recent migrants.

Healthcare inequities among the migrant population have also been described in other countries in Latin America. For instance, in Peru, a high proportion of Venezuelan migrants with self-reported chronic diseases were not receiving treatment, and among those who did, 11.5% did not receive it frequently enough. Moreover, among all migrants seeking medical care, more than half preferred pharmacy attention, self-medication, and primary health care. However, a low proportion reported having experienced discrimination in health facilities [50]. Another study exploring health care access among migrants in transit from Honduras, El Salvador, and Guatemala revealed they seldom used the public health system. When medical attention was needed, 25.9% could not receive timely attention and those who did report going to an informal health care facility or a private hospital. Diverse barriers were identified, as migrants often reported a lack of information since they usually were unaware of where to ask for attention or the facility location, were avoiding the police, did not have financial resources, or were discriminated against because of their migratory status [51]. Another source of inequities is the implementation gap of health-related policies and access to the health care system in countries, such as Colombia and Mexico, which recently started receiving growing intra-regional immigration fluxes, and migratory status ends up conditioning access [52]. Similarly, the bureaucratic process, high cost, and poor intersectoral coordination in Costa Rica have led to regularization difficulties for Nicaraguan migrants, thus impeding them from obtaining health insurance. Additionally, when insured, they report facing exclusion and xenophobia [53]. Similar issues have been identified by medical providers in Uruguay, reporting poor knowledge of the regulatory framework, administrative barriers, differential treatment, and lack of intercultural competence when Afro-Caribbean migrant women need medical attention [54]. Lastly, during the COVID-19 pandemic, inequities have been increasing in Latin America. Migrants have faced difficulties in obtaining timely information, lack of tailored interventions, continuity of care, and barriers to preventive measures [55]. Therefore, the design and implementation of evidence-based health policies with a regional perspective are needed in response to the current inequities observed in different countries in the region.

### 4.3. Limitations and Strengths

There are limitations related to the CASEN survey sampling since it was not designed to be representative of the international migrant population or the population by country of origin. The sample sizes reflect the migratory fluxes observed during the period in which the survey was conducted. According to census data, there was a steady annual increase in the migrant population between 2010 and 2017, and 66.7% of the total migrant population residing in Chile arrived during these years. Specifically, there was a switch in the migratory fluxes from 2017 as a result of political and economic crises in countries of the region (e.g., Venezuela and Colombia), leading to changes in the composition of the migrant population [7] and in the distribution of the studied subgroups that could not be prevented. In addition, sample sizes of emerging recent migrants could have been influenced by their migratory status, social vulnerability, and integration levels in the host country. Some migrants may have refused to participate, did not report that they were born abroad, or were excluded due to a sample design excluding hard-to-reach boroughs. This study has an exploratory and descriptive scope. Furthermore, migratory-related variables such as place of birth have non-response rates ranging from 0.5% to 1.2% each year, resulting in sample sizes that might not be suitable for stratified analysis by demographically and socioeconomically relevant variables for migrant populations, among others. On the other hand, carrying out a secondary analysis implies that limited variables are available for an in-depth understanding of the phenomenon. The inclusion criteria were restricted to residents of private homes, excluding institutionalized migrants and those without permanent housing or facing severe social vulnerability. The survey was conducted in Spanish, in cases where the informant spoke another language the survey feasibility depends on the adequate command of the language. Therefore, some migrants were excluded due to this inherent barrier of the CASEN methodology.

The CASEN 2020 survey released during the socio-sanitary crisis has a different methodological approach. Therefore, we did not include it in our analyses. The data were collected remotely by phone call and the questions wording and structure were modified to facilitate the procedures. The health module was adjusted to the potential impact of COVID-19 pandemic on health care attentions and expected barriers, and some variables were not included, which challenged the comparison with previous CASEN versions.

Despite these limitations, to the best of our knowledge, this is the first study exploring disparities in access to health care services within international migrant groups, specifically emerging and settled migrants in Chile. We recognize that it is descriptive and exploratory. Nevertheless, this methodology provides meaningful insights into health access and use disparities during a time frame when migration was increasing and there were changes in the composition of the migrant population in Chile. Specifically, this analysis puts a special focus on no consultation and lack of treatment coverage due to voluntary and involuntary reasons stratified by insurance type, besides the stratified analysis by demographic characteristics (age, sex) according to the insurance, consultation, and coverage status for each population. This analysis gives an insight into the patterns of health care access and its evolution over time, contributing to the practical knowledge of policies implemented during the studied period. The population-based approach of the CASEN survey allows us to characterize the situation of international migrants residing in Chile and to identify potential disparities that should be further explored and targeted to leave no one behind. It also brings quantitative evidence on specific needs and challenges faced by migrant groups representing the complex migratory flows of Chile and within the Latin-American region, which is relevant for collaborative decision-making. In addition, disaggregated data lead to potential mechanisms that could be further studied.

Access to health care is considered a social determinant of health [56], which can be in itself related to other determinants, such as demographic and socioeconomic factors [23,43]. Thus, the in-depth analysis of these factors and their influence on socially vulnerable groups would shed light on the possible mechanisms behind non-consultation and lack of coverage. The data reported in the present study provide valuable evidence and emerging knowledge of the implementation of current policies to promote access to health care as a goal for global health. It is also relevant for evaluating and designing evidence-based migration policies to reduce gaps in universal and equitable access to health in Chile and throughout the region [57].

### 4.4. Implications in Public Health, Health Services, and Suggestions for Future Research

Broader factors and dynamics are also at play regarding the inequities revealed both by the results of the present studies and by previous studies carried out on access and use of healthcare among international migrants in Chile. First, Chile is characterized by a strong private sector that has progressively been handed the provision of essential social protection services, such as pensions, education, and health, with the State providing mainly for the most vulnerable population groups [58]. More specifically, the overall healthcare system in Chile is fragmented between the private and public systems for insurance coverage and the provision of healthcare services. On the one hand, the private system operates for profit and can refuse coverage to patients whose health risks exceed their payment capacity. Seeking healthcare in private facilities usually carries high out-of-pocket costs that few can afford. On the other hand, the public system offers coverage regardless of personal health and socioeconomic conditions, leading to demand exceeding capacity [59,60,61,62]. Second, migration fluxes, the management of migration, and discourses surrounding migration have changed in the last decades. Recently arrived emerging migrants arguably experience patterns of forced migration linked to socioeconomic, political, and environmental crises in their country of origin, while Chile has been tightening border controls and requirements to obtain visas [63,64,65] leading to increased situations of migratory and socioeconomic precariousness and dynamics of exclusion, including in the healthcare system. Finally, migration has been increasingly politicized, with negative discourses gaining prominence in the public arena [66]. These elements have been proven to impact health care-seeking behaviors in other countries [67,68,69,70,71], something which may be reproduced in Chile, although further specific studies should be carried out to see if this is the case.

In this broader context, profound structural changes are required within the healthcare system, but also in migration governance. Considering that these imply complex political processes which are unlikely to be carried out in the short or medium term, specific, actionable recommendations to improve healthcare access and use among international migrants are needed:Monitor the effective implementation of Supreme Decree no. 67 (*Decreto Supremo no. 67*) in public healthcare centers.Train healthcare workers and administrative staff on migrants’ right to access healthcare and on cross-cultural skills.Provide clear, culturally and linguistically adequate information on the right to health of international migrants regardless of migratory status to migrant communities and how to navigate the healthcare system.Design, pilot, and implement specific programs at the local level to address the challenges faced by recently arrived emerging migrants from an intersectoral perspective.

Diverse initiatives and efforts conducted by the Chilean government to ensure health care access should be sustained, strengthened, and expanded to bring migrants closer to health care providers. Similarly, studies should be conducted regularly to detect challenges faced locally. These initiatives must consider the specific characteristics of the migrant groups residing in Chile, their heterogeneity, and population dynamics, which might improve the understanding of barriers to health care access and use to design appropriate responses. Furthermore, the evolution of regulatory instruments has led to the progressive expansion of migrants’ rights to social protection services, such as maternal care and childcare, immunizations, emergency care, and job security, among others, while facing the challenge of being relevant to refugees, victims of human trafficking, smuggled migrants, migrant children, and irregular migrants [72]. Additionally, it is necessary to comprehensively analyze the impact of the International Migrant Health Policy launched in October 2017 and ongoing international regulation and recommendations ratified in the Latin American region. Therefore, both local and regional collaborative initiatives promoting research and policymaking are essential to foster the social protection of diverse migrant groups currently living in Chile.

In addition, further research should focus on these underserved groups using specific methodological strategies for hard-to-reach populations. The latter could be based on structured questionaries, such as the CASEN survey, and enhanced with specific migratory data for evidence-informed health priority settings among vulnerable groups, including children and adolescent migrants. Finally, future approaches should consider the heterogeneity and dynamics of the population at both local and Latin American regional levels.

## 5. Conclusions

This study analyzed access and use of health care services among the international migrant population in Chile, specifically among settled migrants (from Peru, Argentina, Bolivia, Ecuador), recent emerging migrants (from Venezuela, Dominican Republic, Colombia, Haiti), migrants from other countries, and the Chilean-born population. We found that among immigrants, the percentage of insured population was significantly higher than among the locals. On average, the recent emerging migrant group had a 7.99 times greater uninsured population than the Chilean-born, whereas settled migrants had a 4.69 times greater uninsured population than the locals. Furthermore, compared to 2013, settled and recent emerging migrants showed higher percentages of non-consultation, especially among recent emerging migrants. These findings suggest that different groups of migrants face different barriers and challenges to accessing and using health care systems and services. This phenomenon was exacerbated among recent emerging migrants since many were uninsured and experienced low treatment coverage for diseases included in the Universal Access of Explicit Health Guarantees Plan (AUGE-GES). Regarding recent migrants, it should be considered that this population preferred other types of treatment or got coverage from other sources. However, these exploratory results should be investigated in further detail. In Chile, some gaps have been reduced over time, and sustainable efforts to ensure access and use of health care services have been regionally acknowledged. However, intersectoral solutions are needed to reduce health disparities among locals and within immigrant groups from a human-rights and equity perspective.

## Figures and Tables

**Figure 1 ijerph-20-00741-f001:**
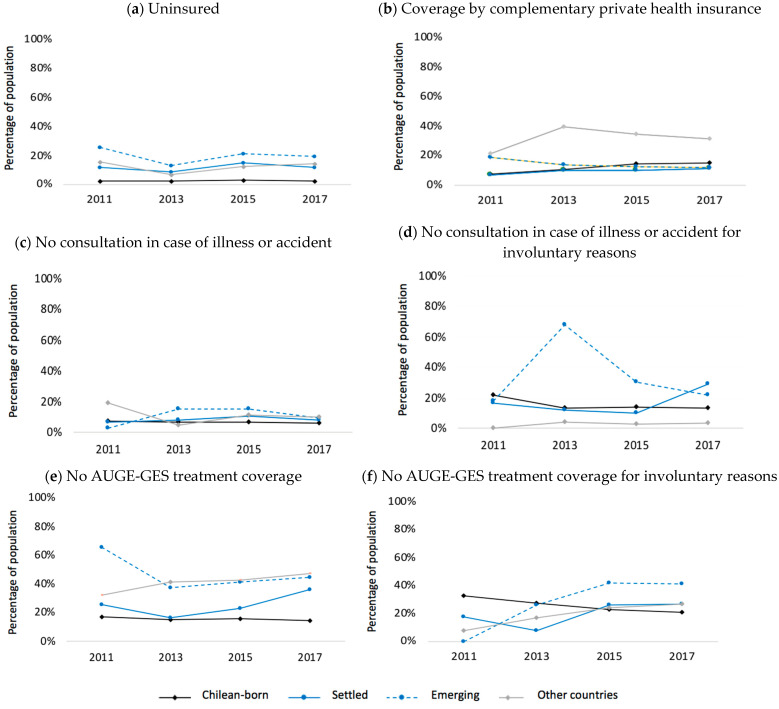
Indicators of access and use of health care services in Chilean-born population and immigrant population (settled, emerging, and other countries). Chile 2011–2017. Settled: Peru, Argentina, Bolivia, Ecuador. Emerging: Venezuela, Dominican Republic, Colombia, Haiti. Others: Migrants from other countries: remaining countries and does not know, does not respond to country of origin. Percentage of: (**a**,**b**) total population, (**c**) population with illness or accident during the three months prior to the survey, (**d**) population that did not consult in case of illness or accident, (**e**) population under AUGE-GES treatment (21 consulted diseases), (**f**) population without AUGE-GES coverage. Chi-2 test, *p*-value: Migration-uninsured: 2011 (0.0366), 2013 (0.0000), 2015 (0.0464), and 2017 (0.0014). Migration-insured: 2011 (0.0000), 2013 (0.0000), 2015 (0.0000), and 2017 (0.0000). Migration-No consultation: 2011 (0.0159), 2013 (0.4425), 2015 (0.7467), and 2017 (0.4700). Migration-No consultation for involuntary reason: 2011 (0.0927), 2013 (0.0000), 2015 (0.0910), and 2017 (0.0536). Migration-no coverage: 2011 (0.3118), 2013 (0.0398), 2015 (0.0322), and 2017 (0.3808). Migration-Non-coverage for involuntary reason: 2011 (0.1869), 2013 (0.0100), 2015 (0.4114), and 2017 (0.0114).

**Figure 2 ijerph-20-00741-f002:**
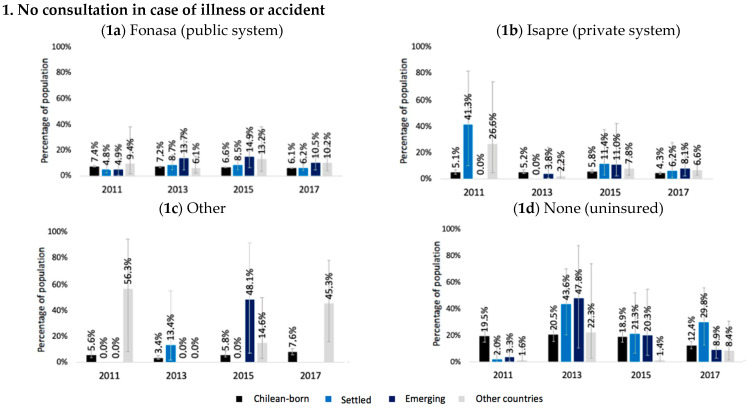
Unexpressed demand: No consultation in case of illness or accident during the three months before the survey and non-AUGE-GES coverage in the Chilean-born population and immigrant population (settled, emerging, and other countries). Chile 2011–2017. Settled: Peru, Argentina, Bolivia, Ecuador. Emerging: Venezuela, Dominican Republic, Colombia, Haiti. Others: Migrants from other countries: remaining countries and does not know, does not respond to country of origin. 1. Percentage of the total population with illness or accident during the three months before the survey. 2. Percentage of the total population on treatment due to any AUGE-GES disease (12 months prior to the survey).

**Table 1 ijerph-20-00741-t001:** Sex and age distribution of Chilean-born population and immigrant population (settled, emerging, and migrants from other countries). Chile 2011–2017.

		Age				Sex	
		0–18	19–30	31–65	66 o more	Men	Women
Chilean-born	2011	28.3% (27.8–28.9%) *n* = 4,702,212	20.0% (19.5–20.5%) *n* = 3,308,622	41.4% (41.0–41.9%) *n* = 6,864,780	10.3% (9.8–10.7%) *n* = 1,701,925	47.6% (47.2–48.0%) *n* = 7,895,139	52.4% (52.0–52.8%) *n* = 8,682,400
	2013	27.6% (27.2–27.9%) *n* = 4,599,075	19.2% (18.9–19.5%) *n* = 3,201,882	42.2% (41.8–42.6%) *n* = 7,045,058	11.0% (10.7–11.4%) *n* = 1,843,362	47.4% (47.0–47.7%) *n* = 7,904,803	52.6% (52.3–53.0%) *n* = 8,784,574
	2015	26.9% (26.6–27.2%) *n* = 4,564,618	19.2% (18.9–19.7%) *n* = 3,271,371	42.1% (41.8–42.4%) *n* = 7,143,977	11.7% (11.4–12.0%) *n* = 1,990,095	47.3% (47.1–47.5%) *n* = 8,026,020	52.7% (52.5–53.0%) *n* = 8,944,041
	2017	25.3% (24.9–25.7%) *n* = 4,266,281	18.7% (18.4–19.0%) *n* = 3,151,050	42.8% (42.5–43.1%) *n* = 7,213,101	13.1% (12.8–13.5%) *n* = 2,213,039	47.5% (47.2–47.7%) *n* = 7,995,319	52.5% (52.3–52.8%) *n* = 8,848,152
Settled migrants	2011	19.9% (15.9–24.7%) *n* = 31,326	34.1% (29.0–39.6%) *n* = 53,518	42.6% (36.6–48.8%) *n* = 66,935	3.4% (2.3–4.9%) *n* = 5348	42.1% (38.2–46.2%) *n* = 66,225	57.9% (53.8–61.8%) *n* = 90,902
	2013	16.8% (14.3–19.6%) *n* = 35,918	31.7% (27.9–35.7%) *n* = 67,813	47.3% (43.0–51.7%) *n* = 101,364	4.2% (3.1–5.8%) *n* = 9022	43.3% (39.1–47.9%) *n* = 92,685	56.7% (52.4–60.9%) *n* = 121,432
	2015	17.0% (14.5–19.7%) *n* = 46,111	33.1% (29.3–37.1%) n= 89,908	47.5% (43.3–51.7%) *n* = 128,985	2.5% (1.9–3.3%) *n* = 6779	47.8% (44.8–50.8%) *n* = 129,893	52.2% (49.2–55.2%) *n* = 141,890
	2017	15.5% (13.4–17.9%) *n* = 34,912	29.0% (26.1–32.2%) n = 125,471	52.4% (48.5–56.3%) *n* = 65,131	3.0% (2.2–4.1%) *n* = 26,658	44.6% (42.1–47.2%) *n* = 132,185	55.4% (52.8–57.9%) *n* = 163,937
Emerging migrants	2011	17.9% (13.0–24.2%) *n* = 4992	39.6% (31.5–48.2%) *n* = 11,036	42.1% (33.9–50.8%) *n* = 11,732	0.4% (0.1–1.6%) *n* = 107	47.1% (64–58.1%) *n* = 13,137	52.9% (41.9–63.6%) *n* = 14,730
	2013	13.8% (10.4–18.2%) *n* = 8662	32.7% (26.8–39.1%) *n* = 20,445	52.7% (47.1–58.2%) *n* = 32,961	0.8% (0.2–2.7%) *n* = 510	48.9% (44.8–53.0%) *n* = 30,598	51.1% (47.0–55.2%) *n* = 31,980
	2015	21.1% (17.4–25.2%) *n* = 22,812	30.1% (25.0–35.9%) *n* = 32,665	47.3% (43.9–50.8%) *n* = 51,315	1.5% (0.8–2.9%) *n* = 1603	48.4% (45.0–51.8%) *n* = 52,458	51.6% (48.2–55.0%) *n* = 55,937
	2017	16.4% (14.5–18.4%) *n* = 63,848	44.9% (39.6–50.4%) *n* = 175,522	37.9% (33.1–43.0%) *n* = 148,003	0.7% (0.4–1.6%) *n* = 3115	51.0% (47.7–54.4%) *n* = 199,332	49.0% (45.6–52.3%) *n* = 191,156
Migrants from other countries	2011	22.7% (15.9–31.4%) *n* = 9090	24.6% (19.0–31.3%) *n* = 14,607	42.9% (34.2–52.0%) *n* = 6047	9.7% (6.5–14.3%) *n* = 1927	50.1% (43.0–57.3%) *n* = 29,516	49.9% (42.7–57.0%) *n* = 29,368
	2013	25.1% (11.6–46.0%) *n* = 19,522	19.3% (14.0–26.0%) *n* = 15,014	45.4% (33.8–57.6%) *n* = 35,386	10.2% (6.9–14.9%) *n* = 7964	46.3% (34.4–58.7%) *n* = 36,070	53.7% (41.3–65.6%) *n* = 41,816
	2015	18.8% (15.5–22.6%) *n* = 16,017	23.4% (18.3–29.3%) *n* = 19,885	49.3% (43.7–55.0%) *n* = 41,998	8.5% (6.0–11.9%) *n* = 7241	48.5% (43.5–53.5%) *n* = 41,265	51.5% (46.5–56.5%) *n* = 43,876
	2017	12.6% (9.7–16.3%) *n* = 11,455	25.8% (21.1–31.1%) *n* = 23,414	50.7% (45.7–55.7%) *n* = 46,045	10.9% (8.6–13.7%) *n* = 9883	51.2% (47.0–55.4%) *n* = 46,522	48.8% (44.6–53.0%) *n* = 44,275

Settled: Peru, Argentina, Bolivia, Ecuador. Emerging: Venezuela, Dominican Republic, Colombia, Haiti. Migrants from other countries: remaining countries and does not know, does not respond to country of origin. ( ): confidence interval, 95% confidence. Chi-2 test, *p*-value. Migration-Sex: 2011 (0.1804), 2013 (0.4452), 2015 (0.9385), and 2017 (0.0008). Migration-age: 2011 (0.0005), 2013 (0.0038), 2015 (0.0000), 2017 (0.0000).

**Table 2 ijerph-20-00741-t002:** Health care insurance in the Chilean-born population and immigrant populations (settled, emerging, and other countries). Chile 2011–2017.

		None (Particular)	Fonasa A.B	Fonasa C.D	Fonasa Does Not Know	Isapre	Other
Chilean-born	2011	2.40% (2.2–2.6%) *n* = 397,011	59.34% (58.1–61.0%) *n* = 9,837,628	18.38% (17.6–19.2%) *n* = 3,046,172	3.58% (3.1–4.1%) *n* = 593,233	12.80% (11.9–13.8%) *n* = 2,122,542	2.48% (2.2–2.7%) *n* = 410,224
	2013	2.53% (2.3–2.8%) *n* = 422,224	53.13% (52.2–54.1%) *n* = 8,867,072	21.02% (20.5–21.6%) *n* = 3,508,536	4.44% (4.2–4.7%) *n* = 740,903	14.15% (13.4–15.0%) *n* = 2,361,099	3.00% (2.8–3.2%) *n* = 493,162
	2015	2.71% (2.6–2.8%) *n* = 459,799	50.71% (49.9–51.5%) *n* = 8,605,778	22.58% (22.2–23.0%) *n* = 3,832,521	4.43% (4.2–4.7%) *n* = 750,845	14.98% (14.3–15.7%) *n* = 2,542,521	2.90% (2.6–3.2%) *n* = 492,232
	2017	2.25% (2.1–2.4%) *n* = 378,239	51.21% (50.4–52.0%) *n* = 8.626,019	22.24% (21.8–22.7%) *n* = 3,746,202	5.20% (4.9–5.5%) *n* = 875,915	14.37% (13.6–15.1%) *n* = 2,419,529	2.83% (2.6–3.1%) *n* = 476,681
Settled migrants	2011	11.43% (8.7–15.0%) *n* = 17,957	53.78% (47.9–59.6%) *n* = 84,498	19.15% (14.4–25.0%) *n* = 30,083	3.42% (2.1–5.6%) *n* = 5371	8.97% (5.8–13.6%) *n* = 14,092	1.287% (0.7–2.4%) *n* = 2022
	2013	8.50% (6.5–11.0%) *n* = 18,207	45.36% (39.7–51.1%) *n* = 97,112	25.87% (21.0–31.4%) *n* = 55,392	7.24% (4.8–10.8%) *n* = 15,508	11.33% (8.3–15.3%) *n* = 24,263	0.723% (0.4–1.2%) *n* = 1547
	2015	14.66% (10.8–19.5%) *n* = 39,839	40.17% (34.5–46.1%) *n* = 109,185	21.33% (17.3–26.0%) *n* = 57,981	8.52% (6.5–11.2%) *n* = 23,157	12.73% (9.0–17.7%) *n* = 34,603	1.07% (0.7–1.7%) *n* = 2910
	2017	11.79% (10.1–13.7%) *n* = 34,912	42.37% (38.2–46.7%) *n* = 125,471	21.99% (19.4–24.8%) *n* = 65,131	9.00% (7.2–11.3%) *n* = 26,658	10.47% (7.7–14.1%) *n* = 31,012	1.58% (0.9–2.7%) *n* = 4681
Emerging migrants	2011	25.31% (13.7–42.1%) *n* = 7052	26.30% (17.8–37.1%) *n* = 7328	16.96% (10.1–27.1%) *n* = 4725	5.83% (2.7–12.0%) *n* = 1624	23.61% (12.3–40.5%) *n* = 6579	0.24% (0.1–0.8%) *n* = 67
	2013	12.85% (8.8–18.4%) *n* = 8041	28.67% (20.2–39.0%) *n* = 17,939	17.41% (12.0–24.5%) *n* = 10,892	11.18% (6.2–19.3%) *n* = 6995	23.09% (15.4–33.2%) *n* = 14,451	0.98% (0.3–3.2%) *n* = 612
	2015	21.06% (15.1–28.5%) *n* = 22,830	32.79% (25.6–40.9%) *n* = 35,545	18.59% (13.4–25.2%) *n* = 20,150	7.19% (4.6–11.0%) *n* = 7792	12.05% (8.4–17.0%) *n* = 13,066	5.22% (1.9–13.9%) *n* = 5662
	2017	19.27% (15.3– 24.1%) *n* = 75,261	32.07% (26.9–37.7%) *n* = 125,215	22.53% (17.3–28.9%) *n* = 87,975	9.93% (6.4–15.1%) *n* = 38,755	12.24% (8.7–16.9%) *n* = 47,779	0.96% (0.4–2.1%) *n* = 3730
Migrants from other countries	2011	15.44% (9.0–25.3%) *n* = 9090	24.81% (16.9–34.9%) *n* = 14,607	10.27% (6.6–15.6%) *n* = 6047	3.27% (1.7–6.3%) *n* = 1927	41.62% (33.1–50.7%) *n* = 24,506	3.77% (1.9–7.5%) *n* = 2218
	2013	6.79% (4.2–10.7%) *n* = 5287	32.68% (18.7–50.6%) *n* = 25,456	13.30% (9.0–19.3%) *n* = 10,361	5.06% (2.7–9.4%) *n* = 3944	32.59% (23.6–43.1%) *n* = 25,381	7.61% (4.4–12.9%) *n* = 5929
	2015	12.22% (7.6–19.1%) *n* = 10,402	19.64% (16.1–23.7%) *n* = 16,720	16.11% (10.9–23.2%) *n* = 13,716	5.04% (2.9–8.6%) *n* = 4293	40.01% (34.0–46.3%) *n* = 34,064	5.68% (3.5–9.0%) *n* = 4837
	2017	14.14% (9.7–20.1%) *n* = 12,840	27.09% (21.9–33.0%) *n* = 24,592	10.02% (7.8–12.7%) *n* = 9094	3.81% (2.6–5.6%) *n* = 3462	38.82% (33.0–44.9%) *n* = 35,248	4.37% (2.6–7.5%) *n* = 3967

Settled: Peru, Argentina, Bolivia, Ecuador. Emerging: Venezuela, Dominican Republic, Colombia, Haiti. Others: Migrants from other countries: remaining countries and does not know, does not respond to country of origin ( ): confidence interval, 95% confidence. Chi-2 test, *p*-value: Migration-Insurance: 2011 (0.0000), 2013 (0.0000), 2015 (0.0000), and 2017 (0.0000).

**Table 3 ijerph-20-00741-t003:** Indicators of access and use of health care services in the Chilean-born population and immigrant population (settled, emerging, and other countries) stratified by sex. Chile 2011–2017.

		Men						Women					
		(a)	(b)	(c)	(d)	(e)	(f)	(a)	(b)	(c)	(d)	(e)	(f)
Chilean-born	2011	243,223	7,181,391	70,500	14,757	195,966	57,458	153,788	8,030,436	82,679	18,279	285,396	98,666
		3.1%	91.0%	7.9%	20.9%	18.0%	29.3%	1.8%	92.5%	6.8%	22.1%	16.1%	34.6%
	2013	260,277	2,960,170	100,128	13,334	186,824	42,652	161,947	2,277,091	123,701	16,667	262,618	80,826
		3.3%	84.8%	7.6%	13.3%	16.0%	22.8%	1.8%	89.4%	6.5%	13.5%	13.9%	30.8%
	2015	274,672	2,878,251	116,688	14,388	207,285	45,427	185,127	2,371,869	135,587	21,659	281,557	64,632
		3.4%	80.8%	7.6%	12.3%	17.2%	21.9%	2.1%	85.9%	6.0%	16.0%	14.4%	23.0%
	2017	231,957	2,761,155	91,342	11,219	190,963	37,484	146,282	2,490,384	112,188	15,331	259,435	55,801
		2.9%	80.0%	6.6%	12.3%	15.3%	19.6%	1.7%	83.9%	5.7%	13.7%	13.4%	21.5%
Settled migrants	2011	8766	60,451	177	39	322	0	9191	84,066	581	88	2518	503
		13.2%	91.3%	3.8%	22.0%	8.3%	0.0%	10.1%	92.5%	9.7%	15.1%	35.2%	20.0%
	2013	9717	39,563	554	16	985	126	8490	30,726	2100	295	1648	75
		10.5%	84.5%	6.6%	2.9%	21.3%	12.8%	7.0%	89.9%	8.5%	14.0%	14.2%	4.6%
	2015	19,748	59,198	2199	120	2602	346	20,091	47,263	2728	377	2520	989
		15.2%	84.3%	9.9%	5.5%	36.0%	13.3%	14.2%	89.1%	11.2%	13.8%	16.8%	39.2%
	2017	17,345	63,263	1516	309	4957	1270	17,567	53,023	2430	845	2807	822
		13.1%	84.0%	9.0%	20.4%	54.4%	25.6%	10.7%	88.0%	8.2%	34.8%	22.9%	29.3%
Emerging migrants	2011	4045	10,751	69	19	17	0	3007	11,885	36	0	640	0
		30.8%	81.8%	13.7%	27.5%	17.5%	0.0%	20.4%	80.7%	1.1%	0	71.0%	0.0%
	2013	3814	15,168	197	0	660	0	4227	7072	1076	860	806	384
		12.5%	83.9%	6.0%	0.0%	75.5%	0.0%	13.2%	82.6%	21.4%	79.9%	26.3%	47.6%
	2015	10,397	27,324	837	0	650	505	12,433	12,953	1341	659	2089	644
		19.8%	87.3%	12.4%	0.0%	48.3%	77.7%	22.2%	78.3%	18.0%	49.1%	39.8%	30.8%
	2017	37,353	99,335	2460	47	1896	902	37,908	59,598	2512	1027	2387	866
		18.7%	82.9%	9.5%	1.9%	45.6%	47.6%	19.8%	87.5%	9.6%	40.9%	43.8%	36.3%
Migrants from other countries	2011	5644	20,455	412	0	969	178	3446	24,378	1090	0	1773	37
		19.1%	69.3%	15.1%	0.0%	20.0%	18.4%	11.7%	83.0%	22.2%	0	48.6%	2.1%
	2013	2307	10,999	207	0	1371	100	2980	6997	357	23	1424	371
		6.4%	59.5%	4.0%	0.0%	42.7%	7.3%	7.1%	59.4%	5.6%	6.4%	39.5%	26.1%
	2015	5277	16,129	1177	0	2697	1003	5125	6374	725	53	1847	88
		12.8%	65.6%	14.8%	0.0%	48.8%	37.2%	11.7%	56.2%	8.0%	7.3%	36.5%	4.8%
	2017	8178	18,948	980	58	2190	188	4662	8350	822	0	2641	1105
		17.6%	66.2%	12.4%	5.9%	43.7%	8.6%	10.5%	58.0%	8.4%	0.0%	50.7%	41.8%

Settled: Peru, Argentina, Bolivia, Ecuador. Emerging: Venezuela, Dominican Republic, Colombia, Haiti. Others: Migrants from other countries: remaining countries and does not know, does not respond to country of origin (a) uninsured population. (b) Population from households in which no member was covered by complementary private health insurance. (c) Population that did not consult in case of illness or accident. (d) Population that did not consult in case of illness or accident for involuntary reasons. (e) Population without AUGE-GES coverage for disease treatment. (f) Population without AUGE-GES coverage for involuntary reasons.

## Data Availability

Data used for this analysis is publicly available at: http://observatorio.ministeriodesarrollosocial.gob.cl/encuesta-casen, accessed on 8 March 2022.

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
