# Peer review of "Unequal Access and Use of Health Care Services among Settled Immigrants, Recent Immigrants, and Locals: A Comparative Analysis of a Nationally Representative Survey in Chile"

_ijerph, 2022, doi:10.3390/ijerph20010741_

Round 1

Reviewer 1 Report (Previous Reviewer 1)

The authors have made significant changes to the manuscript and it is now clearer and more concise. The contribution of the paper to the literature is also evident. There are some minor errors as noted below: 

Abstract: The formatting is confusing. There are numbers in brackets as well as apparent subheadings (Methods, Results, Conclusions) separate by semicolons. Is that in keeping with the journal format? The Conclusion is presented as one long sentence. Can this be rewritten as two shorter sentences for clarity? The word immigrant is used twice in one sentence.

Lines 90-95 are one sentence. Can this be rewritten as two shorter sentences for clarity?

Line 100: do you mean duplicated or doubled?

Line 110: Include citation for Andersen’s framework.

Line 135: Wording is unclear migrants from Argentina had higher odds of presenting chronic conditions,  

Line 145: use of the term “emerging migrants” in this sentence is unclear. Do you mean groups of migrants from specific places or individuals who are just emerging from their own migrant journey?

Author Response

Revisor 1

The authors have made significant changes to the manuscript and it is now clearer and more concise. The contribution of the paper to the literature is also evident. There are some minor errors as noted below.

Author response: thank you for your valuable feedback, we revised the manuscript to address these minor errors and improve the draft’s quality.

  1. Abstract: The formatting is confusing. There are numbers in brackets as well as apparent subheadings (Methods, Results, Conclusions) separate by semicolons. Is that in keeping with the journal format? The Conclusion is presented as one long sentence. Can this be rewritten as two shorter sentences for clarity? The word immigrant is used twice in one sentence.

Author response: Thank you for these suggestions. We accordingly revised the abstract to modify the numbers in brackets and removed subheading following the journal instructions. Moreover, we revised the conclusion for clarity as follows:

“…Globally and particularly in the Latin American region international migration continues to grow. Access and use of health care services by migrants vary according to their country of origin and residence time. We aimed to compare the access and use of health care services between international migrants (including settled migrants from Peru, Argentina, Bolivia, Ecuador; Emerging migrants from Venezuela, Dominican Republic, Colombia, Haiti; and Migrants from other countries) and the Chilean population. Secondary data analysis of population-based nationally representative surveys (CASEN 2011- 2017). Access and use patterns (insurance, complementary insurance, non-consultation, and non-treatment coverage) were described and compared among settled immigrants, recent emerging immigrants, others, and locals. Immigrants had a significantly higher uninsured population compared to locals. Specifically, in CASEN 2017, 19.27% of emerging (95% CI: 15.3%-24.1%), 11.79% of settled (95% CI: 10.1%-13.7%), 2.25% of locals (95% CI: 2.1%-2.4%) were uninsured. Since 2013, settled and recent emerging migrants showed higher percentages of non-consultation. Collaborative and interculturally relevant strategies from human rights and equity perspectives are needed. Initiatives with a particular focus on recent immigrants can contribute to reducing the existing disparities in health care access and use with locals due to lack of insurance and treatment coverage…”

  1. Lines 90-95 are one sentence. Can this be rewritten as two shorter sentences for clarity?

Author response: thank you for this recommendation. We revised these lines as requested:

“…It includes specific pathways to address the health of international migrants, which are complementary to the existing legal framework promoting access to health care services regardless of migratory status. These measures tend to ensure access to emergency attention, sexual and reproductive health, communicable diseases and non-communicable prevention, and pediatric and maternal care, with the overarching goal of achieving the healthiest life possible…”

  1. Line 100: do you mean duplicated or doubled?

Author response: thank you for pointing this out, we mean doubled. We revised as follows:

“…doubled compared to 2015, and utilization of family planning increased by almost 80%..”

  1. Line 110: Include citation for Andersen’s framework.

Author response: thank you for this suggestion. We included the citation as follows.

“…An integrated analytical framework based on Andersen’s framework [20]…”

  1. Line 135: Wording is unclear migrants from Argentina had higher odds of presenting chronic conditions,

Author response: thank for highlighting the need of clarification. We revised as follows:

“…Additionally, migrants from Argentina had higher odds of having chronic conditions, while migrants from Haiti showed higher risks of negative self-perceived health and disability…”

Line 145: use of the term “emerging migrants” in this sentence is unclear. Do you mean groups of migrants from specific places or individuals who are just emerging from their own migrant journey?

Author response: Thank you, we acknowledged that these terms might be unclear. We mean emerging migrant population from Venezuela, Dominican Republic, Colombia, Haiti, whose influx has increased during the last decade. We clarified as follows:

“…The evidence on recent emerging migrants from Venezuela, Dominican Republic, Colombia and Haiti remains scarce. However, it has revealed the challenges faced by Haitian migrants, including a lack of knowledge of available health care services and multiple barriers related to transportation, time availability, and communication…”

Reviewer 2 Report (Previous Reviewer 3)

The revised manuscript has been significantly improved, reaching the basic standard for publication in my opinion. But some details in the manuscript still need attention of authors. For example:

1. The abstract needs to correspond to the revised content in the text. The content in lines 20-24 is obviously different from the description in lines 172-174;

2. Serial number marks in line 112 need reconsideration;

3. The full stop is missing, such as at the end of lines 193 and 567;

4. In line 536 the word seems to be "Afro-Caribbean".

Author Response

Revisor 2

The revised manuscript has been significantly improved, reaching the basic standard for publication in my opinion. But some details in the manuscript still need attention of authors. For example:

Author response: thank you, we appreciate your comment, and we are glad that the manuscript has been significantly improved. We carefully revise these details and improve the quality of the document.

  1. The abstract needs to correspond to the revised content in the text. The content in lines 20-24 is obviously different from the description in lines 172-174.

Author response: Thank you for pointing this out. We acknowledged that the abstract need to correspond to the revised content. We modified as follows:

“…We aimed to compare the access and use of health care services between international migrants (including settled migrants from Peru, Argentina, Bolivia, Ecuador; Emerging migrants from Venezuela, Dominican Republic, Colombia, Haiti; and Migrants from other countries) and the Chilean population…”

  1. Serial number marks in line 112 need reconsideration:

Author response: thank you for this suggestion. We reconsider the number marks as follows:

“… need for health care, resources, predisposing factors and contextual conditions…”

  1. The full stop is missing, such as at the end of lines 193 and 567:

Author response: thank you for spotting our mistake. We accordingly revised as follows:

“…rigorous protocol provided by the Ministry of Social Development. …”

“…were excluded due to this inherent barrier of the CASEN methodology. …”

  1. In line 536 the word seems to be "Afro-Caribbean".

Author response: Thank for your suggestion, we revised as requested:

“…when Afro-Caribbean migrant women need…”

Reviewer 3 Report (Previous Reviewer 4)

Dear authors,

thank you for revising the manuscript. It was very much improved. However, I still think that table 1 was difficult to interprete.

best regards,

Author Response

Revisor 3

Dear authors, thank you for revising the manuscript. It was very much improved.

Author response: We appreciate your feedback, we grateful for your valuable comments to improve the manuscript.

  1. However, I still think that table 1 was difficult to interpret.

Author response: Thank you for highlighting this issue. We propose another format to facilitate the interpretation, in which sex and age distribution is presented separately. Migrant groups and Chilean born are displayed in columns stratified according to each age and sex category. This format shows migratory trends and composition of migrant groups over the years.   We include the new format in this response letter; however, we keep the original table in the revised manuscript for editor’s decision.

Age

0–18

19–30

31–65

66 o more

Year

CB

SM

EM

MO

CB

SM

EM

MO

CB

SM

EM

MO

CB

SM

EM

MO

2011

28.3%
(27.8% - 28.9%)
n = 4.702.212

19.9%
(15.9% - 24.7%)
n = 31.326

17.9%
(13.0% - 24.2%)
n = 4.992

22.7%
(15.9% - 31.4%)
n = 9.090

20.0%
(19.5% - 20.5%)
n = 3.308.622

34.1%
(29.0% - 39.6%)
n = 53.518

39.6%
(31.5% - 48.2%)
n = 11.036

24.6%
(19.0% - 31.3%)
n = 14.607

41.4%
(41.0% - 41.9%)
n = 6.864.780

42.6%
(36.6% - 48.8%)
n = 66.935

42.1%
(33.9% - 50.8%)
n = 11.732

42.9%
(34.2% - 52.0%)
n = 6.047

10.3%
(9.8% - 10.7%)
n = 1.701.925

3.4%
(2.3% - 4.9%)
n = 5.348

0.4%
(0.1% - 1.6%)
n = 107

9.7%
(6.5% - 14.3%)
n = 1.927

2013

27.6%
(27.2% - 27.9%)
n = 4.599.075

16.8%
(14.3% - 19.6%)
n = 35.918

13.8%
(10.4% - 18.2%)
n = 8.662

25.1%
(11.6% - 46.0%)
n = 19.522

19.2%
(18.9% - 19.5%)
n = 3.201.882

31.7%
(27.9% - 35.7%)
n = 67.813

32.7%
(26.8% - 39.1%)
n = 20.445

19.3%
(14.0% - 26.0%)
n = 15.014

42.2%
(41.8% - 42.6%)
n = 7.045.058

47.3%
(43.0% - 51.7%)
n = 101.364

52.7%
(47.1% - 58.2%)
n = 32.961

45.4%
(33.8% - 57.6%)
n = 35.386

11.0%
(10.7% - 11.4%)
n = 1.843.362

4.2%
(3.1% - 5.8%)
n = 9.022

0.8%
(0.2% - 2.7%)
n = 510

10.2%
(6.9% - 14.9%)
n = 7.964

2015

26.9%
(26.6% - 27.2%)
n = 4.564.618

17.0%
(14.5% - 19.7%)
n = 46.111

21.1%
(17.4% - 25.2%)
n = 22.812

18.8%
(15.5% - 22.6%)
n = 16.017

19.2%
(18.9% - 19.7%)
n = 3.271.371

33.1%
(29.3% - 37.1%)
n= 89.908

30.1%
(25.0% - 35.9%)
n = 32.665

23.4%
(18.3% - 29.3%)
n = 19.885

42.1%
(41.8% - 42.4%)
n = 7.143.977

47.5%
(43.3% - 51.7%)
n = 128.985

47.3%
(43.9% - 50.8%)
n = 51.315

49.3%
(43.7% - 55.0%)
n = 41.998

11.7%
(11.4% - 12.0%)
n = 1.990.095

2.5%
(1.9% - 3.3%)
n = 6.779

1.5%
(0.8% - 2.9%)
n = 1.603

8.5%
(6.0% - 11.9%)
n = 7.241

2017

25.3%
(24.9% - 25.7%)
n = 4.266.281

15.5%
(13.4% - 17.9%)
n = 34.912

16.4%
(14.5%- 18.4%)
n = 63.848

12.6%
(9.7% - 16.3%)
n = 11.455

18.7%
(18.4% - 19.0%)
n = 3.151.050

29.0%
(26.1% - 32.2%)
n =125.471

44.9%
(39.6% - 50.4%)
n = 175.522

25.8%
(21.1% - 31.1%)
n = 23.414

42.8%
(42.5% - 43.1%)
n = 7.213.101

52.4%
(48.5% - 56.3%)
n = 65.131

37.9%
(33.1% - 43.0%)
n = 148.003

50.7%
(45.7% - 55.7%)
n = 46.045

13.1%
(12.8% - 13.5%)
n = 2.213.039

3.0%
(2.2% - 4.1%)
n = 26.658

0.7%
(0.4% - 1.6%)
n = 3.115

10.9%
(8.6% - 13.7%)
n = 9.883

Sex

Men

Women

Year

CB

SM

EM

MO

CB

SM

EM

MO

2011

47.6%
(47.2% - 48.0%)
n =7.895.139

42.1%
(38.2% - 46.2%)
n = 66.225

47.1%
(64% - 58.1%)
n = 13.137

50.1%
(43.0% - 57.3%)
n = 29.516

52.4%
(52.0% - 52.8%)
n = 8.682.400

57.9%
(53.8% - 61.8%)
n = 90.902

52.9%
(41.9% - 63.6%)
n = 14.730

49.9%
(42.7% - 57.0%)
n = 29.368

2013

47.4%
(47.0% - 47.7%)
n = 7.904.803

43.3%
(39.1% - 47.9%)
n = 92.685

48.9%
(44.8% - 53.0%)
n = 30.598

46.3%
(34.4% - 58.7%)
n = 36.070

52.6%
(52.3% - 53.0%)
n = 8.784.574

56.7%
(52.4% - 60.9%)
n = 121.432

51.1%
(47.0% - 55.2%)
n = 31.980

53.7%
(41.3% - 65.6%)
n = 41.816

2015

47.3%
(47.1% - 47.5%)
n = 8.026.020

47.8%
(44.8% - 50.8%)
n = 129.893

48.4%
(45.0% - 51.8%)
n = 52.458

48.5%
(43.5% - 53.5%)
n = 41.265

52.7%
(52.5% - 53.0%)
n = 8.944.041

52.2%
(49.2% - 55.2%)
n = 141.890

51.6%
(48.2% - 55.0%)
n = 55.937

51.5%
(46.5% - 56.5%)
n = 43.876

2017

47.5%
(47.2% - 47.7%)
n = 7.995.319

44.6%
(42.1% - 47.2%)
n = 132.185

51.0%
(47.7% - 54.4%)
n = 199.332

51.2%
(47.0% - 55.4%)
n = 46.522

52.5%
(52.3% - 52.8%)
n = 8.848.152

55.4%
(52.8% - 57.9%)
n = 163.937

49.0%
(45.6% - 52.3%)
n = 191.156

48.8%
(44.6% - 53.0%)
n = 44.275

  • CB: Chilean Born. SM: settled migrants from Peru, Argentina, Bolivia, Ecuador. EM: emerging migrants from Venezuela, Dominican Republic, Colombia, Haiti. MO: migrants from other countries (remaining countries and does not know, does not respond to country of origin). ( ): confidence interval, 95% confidence. Chi-2 test, p-value. Migration - Sex: 2011 (0.1804), 2013 (0.4452), 2015 (0.9385), and 2017 (0.0008). Migration– age: 2011 (0.0005), 2013 (0.0038), 2015 (0.0000), 2017 (0.0000).

This manuscript is a resubmission of an earlier submission. The following is a list of the peer review reports and author responses from that submission.

Round 1

Reviewer 1 Report

This article addresses and interesting and relevant topic and acknowledges the heterogeneity of migrant groups. However, the data used for the secondary analysis is relatively limited in terms of representativeness of the larger population groups. It is therefore interesting but difficult to make any meaningful generalisations or novel observations. The limitation in terms of the sample size for subgroups is briefly acknowledged but this needs to be considered in further detail (e.g. Page 3 lines 133-134: the numbers of emerging migrants surveyed seem extremely small relative to the sub-populations (e.g. Page 3 lines 133-134: the numbers of emerging migrants surveyed seem extremely small relative to the sub-populations [2011 – 263 out of a population group of 27, 867 is <1%]). Some additional citations are needed and the reference list seems relatively short and lacks significant links to theory/methodology. I am not convinced of the value of all the graphs and tables that are included on pages 5-10. I have made some specific comments and these are listed below: 

Page 2 line 51: use commas instead of full stops within numbers.

Page 2 lines 53-54: provide citations for the sources of data referred to (census, border control etc).

Page 2 line 79: Change “equitably” to “equitable”

Page 2 line 81: Provide citation for WHA761.17

Page 2 lines 94-97: Reword this sentence that states the gap in current knowledge/literature. This may be clearer if it is rewritten as two sentences

Page 2 line 99 -> page 3 line 100-101: This sentence is unclear and not needed. The authors “propose” an analysis and then state the study aims. Remove “propose” as this paper is describing what has been done.

Page 3 line 129: use commas in numbers not full stops

Page 6 lines 202-203: “Compared to locals, settled migrants, recent emerging migrants and migrants from other countries showed a significantly higher percentage of uninsured population.” The wording of this sentence is unclear.

 Page 11 lines 313-316: “Our data suggest the imperative need of continue reducing inequalities with regards to the local population as well as within immigrant groups by acknowledging the diversity and heterogeneity of the constantly changing migrant population residing in Chile.” This sentence is grammatically incorrect and needs to be reworded.

 Page 11 lines 318-331: The writing in this paragraph is less clear and sentences need to be reworded for grammatical correctness and clarity. For example – “As reported in early crossectional data of SARS‐CoV‐2 pandemic in Chile, where recent migrants faced lack of insurance and COVID‐19 guidance”.

 Page 12 line 373: The wording of the sentence needs to be corrected – “Despite of limitations…”

 Page 13: Conclusion paragraph is unclear and some wording needs to be corrected. E.g. (lins 416-417): the current wording implies a comparison between groups but the only group referred to are “immigrants”. 

Author Response

Revisor 1

  1. This article addresses and interesting and relevant topic and acknowledges the heterogeneity of migrant groups. However, the data used for the secondary analysis is relatively limited in terms of representativeness of the larger population groups. It is therefore interesting but difficult to make any meaningful generalisations or novel observations. The limitation in terms of the sample size for subgroups is briefly acknowledged but this needs to be considered in further detail (e.g. Page 3 lines 133-134: the numbers of emerging migrants surveyed seem extremely small relative to the sub-populations (e.g. Page 3 lines 133-134: the numbers of emerging migrants surveyed seem extremely small relative to the sub-populations [2011 – 263 out of a population group of 27, 867 is <1%]).

Author response: thank you for all valuable comments, we acknowledge the limitations of the available data for secondary analysis that influence sample sizes. Despite these limitations, we proposed an exploratory analysis disaggregated by settled and emerging migrants, which makes visible challenges and gaps among these specific groups. Although this difficulties haven been previously suggested in anecdotical evidence, this the first population-based approach in Chile confirming these observations. We further specify sample limitations in discussion section as follows:

The sample sizes reflect the migratory fluxes observed during the period in which the survey was conducted. According to census data, there was a steady annual increase in the migrant population between 2010 and 2017, and 66.7% of the total migrant population residing in Chile arrived during these years. Specifically, there was a switch in the migratory fluxes from 2017 as a result of political and economic crises in countries of the region (e.g., Venezuela and Colombia), leading to changes in the composition of the migrant population (1) and in the distribution of the studied subgroups that could not be prevented. In addition, sample sizes of emerging recent migrants could have been influenced by their migratory status, social vulnerability, and integration levels in the host country. Some migrants may have refused to participate, did not report that they were born abroad, or were excluded due to a sample design excluding hard-to-reach boroughs

  1. Some additional citations are needed and the reference list seems relatively short and lacks significant links to theory/methodology. I am not convinced of the value of all the graphs and tables that are included on pages 5-10. I have made some specific comments and these are listed below

Author response: We revise the list and included literature to strength the theoretical approach and local evidence throughout the manuscript, as shown in the references list at the end of this response letter

  1. Page 2 line 51: use commas instead of full stops within numbers.

Author response: thank you for spotting our mistake. We have changed as follows “ 1,462,104”

  1. Page 2 lines 53-54: provide citations for the sources of data referred to (census, border control etc).

Author response: Thank you for the suggestion, we revise and updated the citation as requested.

  1. Page 2 line 79: Change “equitably” to “equitable”

Author response: Thank you for pointing this out, we revised as suggested.

  1. Page 2 line 81: Provide citation for WHA761.17

Author response: Thank you for the suggestion, we revise and updated the citation as requested.

  1. Page 2 lines 94-97: Reword this sentence that states the gap in current knowledge/literature. This may be clearer if it is rewritten as two sentences

Author response: Thank you, the sentence was rephrased as follows:

“...To date, there is no temporal analysis of access and use of health care services among historically settled and emerging international migrant populations in Chile. A comparison with the local population over time has never been made either…”

  1. Page 2 line 99 -> page 3 line 100-101: This sentence is unclear and not needed. The authors “propose” an analysis and then state the study aims. Remove “propose” as this paper is describing what has been done

Author response: Thank you, the sentence was rephrased as follows: The purpose of the analysis is to detect patterns and specific needs relevant to evidence-based migration policies and intersectoral responses to guarantee social protection.

  1. Page 3 line 129: use commas in numbers not full stops

Author response: thank you for spotting our mistake. We have changed as follows:

“…Chilean-born population there were 196,421 Chilean-born surveyed in 2011, 212,346 in 2013, 260,754 in 2015 and 207,603 in 2017; (ii) among migrants, there were 2,069 settled migrants surveyed in 2011, 2,418 in 2013, 2,935 in 2015 and 3,491 in 2017. Meanwhile, there were 263 emerging migrants surveyed in 2011, 501 in 2013, 902 in 2015, and 2,346 in 2017. The sample sizes of the international migrant population from other countries of origin were as follows: there were 464 migrants from other countries surveyed in 2011, 636 in 2013, 1,014 in 2015 and 974 in 2017. Non-response rates were 0.54% in 2011; 1.19% in 2013; 0.51% in 2015 and 0.94% in 2017…”

  1. Page 6 lines 202-203: “Compared to locals, settled migrants, recent emerging migrants and migrants from other countries showed a significantly higher percentage of uninsured population.” The wording of this sentence is unclear

Author response: Thank you, the sentence was rephrased as follows

“…The percentage of uninsured populations was significantly higher among settled migrants, recent emerging migrants, and migrants from other countries than among the local population…”

.

  1. Page 11 lines 313-316: “Our data suggest the imperative need of continue reducing inequalities with regards to the local population as well as within immigrant groups by acknowledging the diversity and heterogeneity of the constantly changing migrant population residing in Chile.” This sentence is grammatically incorrect and needs to be reworded.

Author response: Thank you, the sentence was rephrased as follows:

“…Our data suggest the imperative need to continue reducing inequalities with regards to the local population and within immigrant groups by acknowledging the diversity and heterogeneity of the constantly changing migrant population residing in Chile…”

  1. Page 11 lines 318-331: The writing in this paragraph is less clear and sentences need to be reworded for grammatical correctness and clarity. For example – ** Alice en proofreading “As reported in early crossectional data of SARS‐CoV‐2 pandemic in Chile, where recent migrants faced lack of insurance and COVID‐19 guidance”.

Author response: Thank you, the paragraph was edited for correctness, clarity and conciseness: 

“…Migration-related variables such as country of origin and residence time shape migration experiences in Latin America and the Caribbean. This is aligned with dynamic population movements in the region, and processes of settlement and integration (33). Regarding more specifically the time of residence in the destination country, there are three critical periods of interest for disease diagnosis and treatment: arrival, recent arrival, or newcomers (without univocal definition, ranging between 6 months and 8 years), and settled migrants. Immediately after arrival, diseases or accidents derived from the migration process may occur. In addition, emerging or recently imported diseases might appear during this period since protection and prevention measures may not have been adequately carried out, for example, during a pandemic or health crisis (34). This was reported in early cross-sectional data on the SARS-CoV-2 pandemic in Chile, where recent migrants faced lack of insurance and COVID-19 guidance (32). Therefore, it is necessary to provide precise orientation on navigating the health system…”

  1. Page 12 line 373: The wording of the sentence needs to be corrected – “Despite of limitations…”

Author response: Thank you, the sentence was rephrased as follows:

“…Despite these limitations, to the best of our knowledge, this is the first study exploring disparities in access to health care services within international migrant groups, specifically emerging and settled migrants in Chile…”

  1. Page 13: Conclusion paragraph is unclear and some wording needs to be corrected. E.g. (lins 416-417): the current wording implies a comparison between groups but the only group referred to are “immigrants”. 

Author response: Thank you, the conclusion paragraph was revised for clarity and correctness and the specific sentence you flagged was reworded as follows:

“… We found that among immigrants, the percentage of insured population was significantly higher than among the locals…”

Reviewer 2 Report

The topic of this article focuses on health inequities among immigrants, and the findings have important policy implications. The article focuses on some descriptive statistics that examine differences in health service utilization across populations. The reviewers believe that the academic value of the article could have been higher if it had provided a more explanatory theoretical framework for statistical inferential analysis of the causes of health service utilization disparities.

For example, the study found that settled migrants (settled migrants) had a higher probability of not participating in health counseling. So, what factors contribute to this phenomenon and what theories can explain it? Perhaps the causal mechanisms behind it can be further explored to give a convincing explanation.

Author Response

Revisor 2

  1. The topic of this article focuses on health inequities among immigrants, and the findings have important policy implications. The article focuses on some descriptive statistics that examine differences in health service utilization across populations. The reviewers believe that the academic value of the article could have been higher if it had provided a more explanatory theoretical framework for statistical inferential analysis of the causes of health service utilization disparities.

Author response: Thank you for your comment, we appreciate this valuable feedback. We hope to provide a better theoretical framework in order to improve the quality of the manuscript. At the introduction section we included a framework to explain potential health service utilization disparities as follows:

“…Despite these valuable efforts, evidence suggests that international migrants and their families have poorer access to the health care system and use health services less frequently when compared to the local population (2). Additionally, migrants are less likely to be insured and use services when compared to the local population (3). Access to health care is a multidimensional definition that implies using the health care system when needed, considering availability and the ability to physically access and afford it (4, 5). An integrated analytical framework based on Andersen’s framework could explain healthcare utilization disparities. The modified model proposes the following categories: i) need for health care; ii) resources; iii) predisposing factors and; iv) contextual conditions. Low levels of health care needs are determined by self-perceived or formally assessed health status, which may differ across gender, ethnical background, and specific needs derived from the migratory process. Furthermore, migrants might have differential resources impacting their access, such as financial resources, social resources (e.g., social capital and support might encourage specific health-seeking behaviors or be less available among recent migrants), and the lack of human resources, devices, health facilities, and interpreters. Specifically, financial and social resources could favor migrants’ transnational access or preference for their traditional medicine or religious counseling, decreasing health care use in the destination country. The latter could also result from mistrust in health care providers in the destination country due to a lack of cross-cultural skills (3). Besides individual factors influencing healthcare utilization (e.g., demographic, socioeconomic), there are broader factors specific to the migrant population. For example, migratory status could lead to fearing deportation, discrimination, or not being entitled or eligible for certain health services. Moreover, the context of migration and reception interacts with macrostructural conditions in the destination country, including political and socioeconomic context and health care system policies and efficiency (3)…”

  1. For example, the study found that settled migrants (settled migrants) had a higher probability of not participating in health counseling. So, what factors contribute to this phenomenon and what theories can explain it? Perhaps the causal mechanisms behind it can be further explored to give a convincing explanation.

Author response: Thank you for pointing this out, the possible causal mechanisms behind the non-consultation should be discussed. It is worth to mention that recent emerging migrants showed higher percentages of non-consultation over the studied period. However, from 2015 non-consultation due to involuntary reasons increased among settled migrants, without significant differences. Firstly, in the discussion section we highlighted the possible factors that influence health care utilization in recent migrants. Subsequently, we discuss potential factors promoting non-consultation by involuntary reasons among settled migrants, including resources and contextual conditions such as

“…Differing health care needs could explain higher percentages of non-consultation among emerging migrants compared to other migrants or the local population. On the one hand, recent migrants are often healthier than the local population. This phenomenon has been described as “the healthy migrant effect” hypothesis, where a health advantage is observed by lower morbidity and mortality rates compared to the native-born population. It is commonly attributed to a self-selection process, where younger, healthier, and wealthier people are more prone to migrating (6). In Chile, previous evidence has reported a possible “healthy migrant effect” on various health indicators such as disability, chronic diseases, accidents, and hospitalization rates (7-11). Furthermore, as previously mentioned, those who have spent more time in the country tend to see their health deteriorate, possibly due to assimilation processes, suggesting the loss of their health advantage over time (12). Therefore, their need for health care is expected to increase with the longer they stay. On the other hand, resources and contextual conditions might also explain access and use disparities. For example, recent migrants are building financial and social resources during their integration process while learning how to navigate an unfamiliar health care system, in some cases experiencing lower language proficiency or poor access to information (3). Particularly, recent emerging migrants might lack social support and help to overcome access barriers, limiting health care-seeking behaviors. In addition, the context of the destination country and the attitudes of the host population could influence health care access and utilization. Specifically, migrants with irregular administrative status face higher vulnerability and often experience administrative, financial, and communicational barriers, as well as not being eligible for social protection measures (13).

Among settled migrants, there was a tendency towards increased non-consultation for involuntary reasons throughout the study period. Although it is expected that settled migrants increase their health care needs over time, some difficulties could not be prevented and might be related to migrant-specific predisposing factors, resources, and contextual factors. Although settled insured migrants are entitled to the same rights as the local population, they tend to use it less even when needed. Migrants in Latin America have faced challenges in using health care services, making them turn to alternative pathways. These challenges include discrimination or differential treatment by medical providers when looking for medical attention. Moreover, this can be reinforced by a lack of awareness of their rights and knowledge of the services to which they are entitled (14). Additionally, there are issues related to the health system, for example, the lack of availability of health care professionals and specialists according to the need of the population, which might exacerbate the exclusion of the migrant population (3), particularly when the needs of settled migrants might be similar to the ones of the local population. On the other hand, working conditions could influence time availability to reach the place of care, time off for medical appointments, and financial resources (15). Although settled migrants have higher levels of integration in Chile, they often work longer hours, have lower wages and poor employment status, hindering social protection (16). However, the possible mechanism behind non-consultation for involuntary reasons in settled migrants should be further studied from an intersectoral perspective to understand better these unexpected disparities with regards to recent migrants…”

Reviewer 3 Report

This study proposes an issue worthy of attention. The paper used secondary data to descriptively analyze the access and use of health care services related to international migrants residing in Chile, which could help improve health care policies for Chile and other developing countries with immigrants. Some comments and suggestions for consideration are as follows.

(1) In the ‘Introduction of this paper, the current health care situation in Chile is introduced, but there seems no review of the existing papers. The authors should illustrate the development situation of theoretical or practical originality in this paper. For example, are there any papers related to the medical situation of different groups of people? What are the limitations of these papers?

(2) It is difficult to find new or original methods proposed by this paper. The authors need to add relevant papers on secondary data analysis in the literature review and expand the analysis methods for the secondary data in this paper. Introducing and describing the healthcare service dilemma in Chile are well done. Unfortunately, the data analysis could not clearly show the original contributions. Although the CASEN survey has a considerable degree of authority and reliability, the analysis approach is a simplistic way (the chi-2 and F test show the sample is persuasive, and then?). The selection of samples directly determines the reliability of the analysis results, so more text is needed to describe the sample selection process. Besides, the issue of inequality in health care resources of immigrants deserves more attention, and I hope for a more in-depth study on this topic. A significant portion of the descriptive statistics in the paper could be presented in graphs rather than text (e.g., lines 124-144)

(3) More specific recommendations should be made based on the conclusions of the analysis, such as public policy recommendations for different government departments. My suggestion is to combine with more to Chile's national conditions, policies, and regulations when interpreting the results. For example, changes in the number of different types of immigration over time, as well as changes in gender ratios, should be explained more in conjunction with the policies and events associated with them.

Author Response

Revisor 3

This study proposes an issue worthy of attention. The paper used secondary data to descriptively analyze the access and use of health care services related to international migrants residing in Chile, which could help improve health care policies for Chile and other developing countries with immigrants. Some comments and suggestions for consideration are as follows.

Author response: Thank you for this comment, we appreciate this feedback. We hope to provide clear revision for each comment

  1. In the ‘Introduction of this paper, the current health care situation in Chile is introduced, but there seems no review of the existing papers. The authors should illustrate the development situation of theoretical or practical originality in this paper. For example, are there any papers related to the medical situation of different groups of people? What are the limitations of these papers?
  2. It is difficult to find new or original methods proposed by this paper. The authors need to add relevant papers on secondary data analysis in the literature review and expand the analysis methods for the secondary data in this paper

Author response: Thank you for highlighting the need of revising the evidence in Chile. We added to the introduction specific details and gaps of the current national literature. In addition, we have accordingly added relevant papers on secondary data analysis and specified the limitation of previous literature supporting the originality of the methods proposed by this paper as follows:

“…Previous population-based data have shown gaps between the Chilean-born population and international migrants on insurance, levels of addressed health care needs, and treatment coverage, highlighting the shortcomings in affiliation to the health care system (17). Meanwhile, another analysis focused on migrants’ health status reported its association with insurance status. Specifically, chronic morbidity was associated with public and private health system affiliation, probably due to higher access to diagnosis and treatment. Additionally, migrants from Argentina had higher odds of presenting chronic conditions, while migrants from Haiti showed higher risks of negative self-perceived health and disability (11). Although these studies bring attention to the issues of access and health risk among migrants in Chile, the data is limited to the CASEN 2017 survey and does not reflect the migratory trends and composition of migrant groups over the years.

Moreover, national literature has described diverse barriers to mental health care (15, 18), child preventive care and checkups to track child growth and development (19), and sexual and reproductive health care for adolescents, among others (20). In addition, primary health care workers have reported difficulties in providing adequate attention to international migrants due to financial resources strains, unclear regulations, lack of appropriate registries, and cultural barriers (21). The evidence on recent emerging migrants remains scarce. However, it has revealed the challenges faced by Haitian migrants, including a lack of knowledge of available health care services and multiple barriers related to transportation, time availability, and communication (22). Specifically, the evidence on maternity of Haitian women describes their difficulties navigating the Chilean health system, not knowing where to get medical attention, and unmet sexual, reproductive, and mental health needs (23). Similarly, the available evidence on recently arrived migrants from Venezuela suggests that a notable proportion of people did not know where to seek medical help during the COVID-19 pandemic and had high levels of non-affiliation (24). Overall, the literature has described existing gaps mainly between the local population and migrants without disaggregating according to international migration patterns over time, including emerging recent migrants and settled migrants. The existing evidence has reported multiple barriers experienced by specific migrant groups without quantifying nor verifying suggested disparities in access and use of health care services from population-based data. Therefore, previous evidence did not capture distinctions or make particular challenges visible among migrant groups and the local population while migratory fluxes were changing…”

  1. Introducing and describing the healthcare service dilemma in Chile are well done. Unfortunately, the data analysis could not clearly show the original contributions. Although the CASEN survey has a considerable degree of authority and reliability, the analysis approach is a simplistic way (the chi-2 and F test show the sample is persuasive, and then?).

Author response: thank you for your constructive commentary. In the discussion we have clarified the contributions of the study. We specified the focus of the analysis and acknowledged its exploratory scope for decision makers, policy implementation evaluation and future research of potential mechanisms behind the pattern of access and use of health care services during a time frame in which critical migratory phenomena were taking place.

“… Despite these limitations, to the best of our knowledge, this is the first study exploring disparities in access to health care services within international migrant groups, specifically emerging and settled migrants in Chile. We recognize that it is descriptive and exploratory; nevertheless, this methodology provides meaningful insights into health access and use disparities during a time frame when migration was increasing and there were changes in the composition of the migrant population in Chile. Specifically, this analysis puts a special focus on no consultation and lack of treatment coverage due to voluntary and involuntary reasons stratified by insurance type, besides the stratified analysis by demographic characteristics (age, sex) according to the insurance, consultation, and coverage status for each population. This analysis gives an insight into the patterns of health care access and its evolution over time, contributing to the practical knowledge of policies implemented during the studied period. The population-based approach of the CASEN survey allows us to characterize the situation of international migrants residing in Chile and to identify potential disparities that should be further explored and targeted to leave no one behind. It also brings quantitative evidence on specific needs and challenges faced by migrant groups representing the complex migratory flows of Chile and within the Latin-American region, which is relevant for collaborative decision-making. In addition, disaggregated data lead to potential mechanisms that could be further studied…”

  1. The selection of samples directly determines the reliability of the analysis results, so more text is needed to describe the sample selection process.

Author response: Thank you for your comment, a more detailed description of selection process was included in methods as follows: 

“…The anonymized data covers diverse topics such as education, work, income, health, and housing, among others. The survey is carried out by the Ministry of Social Development every 2-3 years, based on a probabilistic, stratified, and two-stage sampling representative of Chilean households at national, regional (including 16 regions of the national territory), and urban/rural levels. The sampling design was conducted by the National Institute of Statistics, having sampling frames for urban (called blocks at the commune-area-group size level) and rural areas (called sections at commune-area size level) made up of conglomerates of grouped houses where household members were residing. The urban and rural areas were selected proportional to its size, while houses were systematically selected with equal probability within the selected clusters. The sampling procedures excluded hard-to-reach boroughs and population residing in prisons and health care facilities. For instance, the sample of the CASEN survey 2017 was made up of 70.947 households with 216.439 residents representing 16.843.471 Chilean-born and 777.407 international migrants…”

  1. Besides, the issue of inequality in health care resources of immigrants deserves more attention, and I hope for a more in-depth study on this topic
  2. More specific recommendations should be made based on the conclusions of the analysis, such as public policy recommendations for different government departments. My suggestion is to combine with more to Chile's national conditions, policies, and regulations when interpreting the results. For example, changes in the number of different types of immigration over time, as well as changes in gender ratios, should be explained more in conjunction with the policies and events associated with them.

Author response: thank you for these valuable insights, as suggested we further discusses inequality in health care resources while considering national conditions, policies and regulation. In order to provide evidence-based recommendations as follows:

“…Broader factors and dynamics are also at play regarding the inequities revealed both by the results of the present studies and by previous studies carried out on access and use of healthcare among international migrants in Chile. First, Chile is characterized by a strong private sector that has progressively been handed the provision of essential social protection services such as pensions, education, and health, with the State providing mainly for the most vulnerable population groups(25). More specifically, the overall healthcare system in Chile is fragmented between the private and public systems for insurance coverage and the provision of healthcare services. On the one hand, the private system operates for profit and can refuse coverage to patients whose health risks exceed their payment capacity; seeking healthcare in private facilities usually carries high out-of-pocket costs that few can afford. On the other hand, the public system offers coverage regardless of personal health and socioeconomic conditions, leading to demand exceeding capacity (26-29). Second, migration fluxes, the management of migration, and discourses surrounding migration have changed in the last decades. Recently arrived emerging migrants arguably experience patterns of forced migration linked to socioeconomic, political, and environmental crises in their country of origin, while Chile has been tightening border controls and requirements to obtain visas (30-32) leading to increased situations of migratory and socioeconomic precariousness and dynamics of exclusion, including in the healthcare system. Finally, migration has been increasingly politicized, with negative discourses gaining prominence in the public arena (33). These elements have been proven to impact health care-seeking behaviors in other countries (34-38) something which may be reproduced in Chile, although further specific studies should be carried out to see if it is the case.

In this broader context, profound structural changes are required within the healthcare system but also in migration governance. Considering that these imply complex political processes which are unlikely to be carried out in the short or medium term, specific, actionable recommendations to improve healthcare access and use among international migrants are needed:

  • Monitor the effective implementation of Supreme Decree nº67 (Decreto Supremo nº67) in public healthcare centers.
  • Train healthcare workers and administrative staff on migrants’ right to access healthcare and on cross-cultural skills.
  • Provide clear, culturally, and linguistically adequate information on the right to health of international migrants regardless of migratory status to migrant communities and how to navigate the healthcare system.
  • Design, pilot, and implement specific programs at the local level to address the challenges faced by recently arrived emerging migrants from an intersectoral perspective...”

  1. A significant portion of the descriptive statistics in the paper could be presented in graphs rather than text (e.g., lines 124-144)

Author response: thank you for this suggestion. However, we believe that providing the actual numbers is useful for a broad range of readers and purposes. Since this study shall serve as a reference for policy making, researching and policy implementation evaluation.

Reviewer 4 Report

Dear Authors,

The information in the manuscript has an extremely high value in expressing health equitya . However, I have few point as follow:

In the introduction part, is there any policy implementation that is more rec17? It looks like Chile has many immigrants, and the higher authority widely acknowledges the health inequity (as you mentioned in the introduction). Moreover, if the last policy change were in 2017, it would be very interesting to see the trend of healthcare access and use change after the implementation.

Method: I saw that you have an interview part in collecting data as well; how did you sample the participants for those interview sessions, and how did you collect the data in that part? Furthermore, regarding the  CASEN survey, was everyone in the country responding to the survey, or how did you pick participants from those population groups?

The data was a little old if the analysis and data collection was done in 2017, which was 5 years ago; I believe that many things have changed, especially after the COVID-19 pandemic, which increased the health inequity in both access to care and healthcare utilization.

Please define 'access to healthcare' and the 'use of healthcare' in your primary aim. Which parameters help understand access to care? Which results were the primary outcomes to expect and say that they determined good access to care or the access to care is improving throughout the years?

Since this study included migrants for self-reporting and interviewing, would the language be a barrier to the data collection. Did you determine the language literacy in each immigrant group? Were they properly understand the questions?

Please state the ethical consideration points in the manuscript concerning the interview.

Results: I think the format of table 1 can be improved. Would it be better to change the format to graphs so that we could see the trend of age and gender each year from eachandn group as well as the total immigrant trend in comparison to Chilean-born?

I think the time of residence after the migration is also interesting to show; it should be one of the factors determining healthcare access.

All descriptive data of the participants, like age, sex, number of each group, number, and type of insurance, should be shown in 1 table so that we could picture the whole samples and relate other results that you present later on to these participants' characteristics.

In the paragraph from lines 265-276, you discussed Figure 1a-1d in Figure 2. The graphs' labels were confusing; please change the label in the graphs to match the text.

What did Figure 2a-2d mean? Why people in the public system had Non‐coverage in the case of AUGE‐GES disease? And from the information in the discussion, would it mean that some of the diseases in AUGE‐GES disease are not important to put there since people preferred using other treatment paths?

Discussion: please also try to discuss other countries' healthcare inequity contexts in other studies.

Best regards,

Reviewer

Author Response

Revisor 4

  1. In the introduction part, is there any policy implementation that is more rec17? It looks like Chile has many immigrants, and the higher authority widely acknowledges the health inequity (as you mentioned in the introduction). Moreover, if the last policy change were in 2017, it would be very interesting to see the trend of healthcare access and use change after the implementation.

The National Health Plan for 2020 and its National Health Strategy 2011-2020 state the importance of explicitly considering specific health goals for international migrants (Strategic Line No. 5: Equity and Health) (39). The Pilot Plan for International Migrants' Health was designed in 2015 and executed in 2016-2017, leading to the formulation of the International Migrant Health Policy (40). This policy represents a step towards universal care and adopts intercultural and intersectoral perspectives. It includes specific pathways to address the health of international migrants, which are complementary to the existing legal framework promoting access to health care services regardless of migratory status, among which emergency attention, sexual and reproductive health, communicable diseases and non-communicable prevention, and pediatric and maternal care, with the overarching goal of achieving the healthiest life possible (41). Notably, a legal provision from 2016 granted irregular migrants access to the public health system, and when the International Migrant Health Policy was launched in 2017, the proportion of insured migrants was about twice that reported in 2013. Moreover, compared to evidence from 2015, there was a reduction in perceived difficulties in obtaining a health care appointment, along with higher health care utilization. Specifically, in 2017 use of prenatal care was duplicated compared to 2015, and utilization of family planning increased by almost 80%.  This progress was also observed through hospital discharge rates, where the proportion of uninsured discharged migrants decreased from 25% in 2014 to 7% in 2019 (41)

  1. Method: I saw that you have an interview part in collecting data as well; how did you sample the participants for those interview sessions, and how did you collect the data in that part? Furthermore, regarding the CASEN survey, was everyone in the country responding to the survey, or how did you pick participants from those population groups?

Author response: Thanks for highlighting the need of clarification. The mode of data collection of CASEN survey is personal interview with a household respondent over 18 years. Therefore, there weren’t other interview sessions apart from the survey. Regarding the sampling procedures of CASEN survey, we provided a detailed description of the sampling design which is representative at national, regional and both urban and rural level, including 216.439 household residents (CASEN 2017) throughout the country weighted according to the sample design as follows:

“…The survey is carried out every 2-3 years by the Ministry of Social Development, based on a probabilistic, stratified and two-stage sampling that is representative of overall residents of Chilean households at each national, regional (including 16 regions of the national territory), and urban/rural level. The sampling design was conducted by the National Institute of Statistics, having sampling frames for urban (called blocks at the commune-area-group size level) and rural areas (called sections at commune-area size level) made up of conglomerates of grouped houses where household members were residing. The urban and rural areas were selected proportional to its size, while houses were systematically selected with equal probability within the selected clusters. The sampling procedures excluded hard-to-reach boroughs and population residing in prisons and health care facilities. For instance, the sample of the CASEN survey 2017 was made up of 70.947 households with 216.439 residents representing 16.843.471 Chilean-born and 777.407 international migrants…”

Author response: in addition, we clarified that personal interview was the mode of data collection of the survey conducted by trained pollsters (instead of “interviewers” to avoid misunderstandings). As follows: 

“…Data was collected through personal structured interviews conducted by trained field pollsters throughout the Chilean territory (excluding hard-to-reach areas)…”

  1. The data was a little old if the analysis and data collection was done in 2017, which was 5 years ago; I believe that many things have changed, especially after the COVID-19 pandemic, which increased the health inequity in both access to care and healthcare utilization

Author response: thank you for this comment, we acknowledge the limitations of data up to 2017 without reporting the effect of COVID-19 pandemic. However, this study gives important insights of the situation and trends prior to the sanitary emergency, useful for further comparisons when data in Chile become available. In the discussion we further specified the reasons why we did not include the CASEN 2020, considering the differences in the methodological approach that were not comparable to CASEN 2011-2017.

“…The CASEN 2020 survey released during the socio-sanitary crisis has a different methodological approach, therefore we did not include it in our analyzes. The data was collected remotely by phone call and the questions wording and structure were modified to facilitate the procedures. The health module was adjusted to the potential impact of COVID-19 pandemic on health care attentions and expected barriers, and some variables were not included which difficulted the comparison with previous CASEN versions...”

  1. Please define 'access to healthcare' and the 'use of healthcare' in your primary aim. Which parameters help understand access to care?

Author response: Thank you for this suggestion. We define these terms in the introduction section, in addition we described a theorical framework explaining the access to care in international migrants to better understand potential disparities.

“…Access to health care is a multidimensional definition that implies using the health care system when needed, considering availability and the ability to physically access and afford it (4, 5). An integrated analytical framework based on Andersen’s framework could explain healthcare utilization disparities. The modified model proposes the following categories: i) need for health care; ii) resources; iii) predisposing factors and; iv) contextual conditions. Low levels of health care needs are determined by self-perceived or formally assessed health status, which may differ across gender, ethnical background, and specific needs derived from the migratory process. Furthermore, migrants might have differential resources impacting their access, such as financial resources, social resources (e.g., social capital and support might encourage specific health-seeking behaviors or be less available among recent migrants), and the lack of human resources, devices, health facilities, and interpreters. Specifically, financial and social resources could favor migrants’ transnational access or preference for their traditional medicine or religious counseling, decreasing health care use in the destination country. The latter could also result from mistrust in health care providers in the destination country due to a lack of cross-cultural skills (3). Besides individual factors influencing healthcare utilization (e.g., demographic, socioeconomic), there are broader factors specific to the migrant population. For example, migratory status could lead to fearing deportation, discrimination, or not being entitled or eligible for certain health services. Moreover, the context of migration and reception interacts with macrostructural conditions in the destination country, including political and socioeconomic context and health care system policies and efficiency (3)…”

  1. Which results were the primary outcomes to expect and say that they determined good access to care or the access to care is improving throughout the years?

Author response: Thank you for this comment, the primary outcomes were specified in the methods section. Steady improvements are expected as a result of progressive implementation of existing legal framework during the study period. Including higher proportion of migrants insured, effective medical consultation among those who expressed demand for health care services in case of illness/accident or having treatment coverage for AUGE-GES diseases (Figure 1, 2) . Therefore, a closer gap between local population and international migrants that should be also influenced by changes in the composition of migrant groups due to migratory fluxes. Despite of the expected improvements, the emerging recent migrants might experience lower access and utilization due to their higher vulnerability, lack of information, among others explained in the discussion.

  1. Since this study included migrants for self-reporting and interviewing, would the language be a barrier to the data collection. Did you determine the language literacy in each immigrant group? Were they properly understand the questions?

Author response: Thank you for considering the language barrier that it is worth to point out. The CASEN survey is mainly restricted to Spanish-speaker informants or the pollster speaks the language of the selected informant. Thus, the language literacy was not determined, and some migrant groups were excluded. This limitation was included as follows:

“...The survey was conducted in Spanish, in cases where the informant spoke another language the survey feasibility depends on the adequate command of the language. Therefore, some migrants were excluded due to this inherent barrier of the CASEN methodology…”

  1. Please state the ethical consideration points in the manuscript concerning the interview

Author response: Thank you for this valuable comment. We specified ethical considerations in the methods as follows:

“…Prior to the survey the trained pollster explained the statistical purposes of the data and its confidentiality, specifying the anonymization process and institutional contacts to resolve any question. The informant provides consent and the survey was conducted following rigorous protocol provided by the Ministry of Social Development…”

  1. Results: I think the format of table 1 can be improved. Would it be better to change the format to graphs so that we could see the trend of age and gender each year from eachandn group as well as the total immigrant trend in comparison to Chilean-born?

Author response: Thank you for this suggestion. However, we believe is appropriate to show the actual number as suggested by reviewer 1. In addition, we included as supplementary material a table with graphs to see the trends in each group as a proposal for review (supplementary table 1).

  1. I think the time of residence after the migration is also interesting to show; it should be one of the factors determining healthcare access.

Author response: Thank you, we agree that time of residence is interesting to show. We included a supplementary table reporting the challenges of uninsured migrants and those without AUGE-GES treatment coverage and non-consultation in case of illness or accident (supplementary table 2).

  1. All descriptive data of the participants, like age, sex, number of each group, number, and type of insurance, should be shown in 1 table so that we could picture the whole samples and relate other results that you present later on to these participants' characteristics.

Author response: We appreciate this suggestion; however, we believe that is appropriate to show age and sex distribution separate to prevision to make easier to locate the data among a large number of values in the table 1.

  1. In the paragraph from lines 265-276, you discussed Figure 1a-1d in Figure 2. The graphs' labels were confusing; please change the label in the graphs to match the text.

Author response: thank you for pointing this out, we have accordingly modified the labels as requested.

  1. What did Figure 2a-2d mean? Why people in the public system had Non‐coverage in the case of AUGE‐GES disease? And from the information in the discussion, would it mean that some of the diseases in AUGE‐GES disease are not important to put there since people preferred using other treatment paths

Author response: thanks for highlighting the need of clarification, we acknowledge that AUGE-GES non-coverage among people in the public system and preference for other treatment paths should be further explained. We accordingly specified potential reasons from the evidence of how the AUGE-GES work in the discussion.

“…The AUGE-GES prioritize the health conditions that are most severe, prevalent, expensive, and affecting quality of life, in response to the increase of chronic and degenerative diseases (42). The plan contains administrative conditions that might influence its effective coverage, and access is determined by the health authority protocol, available resources, and capacities of health care services. The “opportunity” guarantee defines a time limit in which the health benefits can be provided, with the approval of a medical specialist. This requirement, in particular, causes long waiting lists since there is a general lack of specialists in the public health system, especially outside of the capital region of Santiago (42, 43). Other key issues influence the quality of these guarantees. For instance, standards are met only by a limited number of providers because continuous training is scarce and devices and human resources are insufficient in the public system (43). These challenges may be worsened by the existent strategies for financial protection where the public system allocates its funding to the private system when it cannot provide treatment, promoting the growth of the private system while restricting funding to improve the public system. In addition, age requirements might impede adequate coverage of services such as surgical treatment, those under or over the requested age are excluded. In contrast, eligible people might also face the difficulties of waiting lists (43). Furthermore, the selection of AUGE-GES diseases has restricted the attention of other health conditions favoring exclusion, segmentation, and the failure to fulfill the right to health among specific population groups by hindering the principle of universality and equity. Along with the mentioned administrative shortcomings, financial constraints due to fee increases or pricing errors (42) might promote preferences for alternative pathways and treatments…”

  1. Discussion: please also try to discuss other countries' healthcare inequity contexts in other studies.

Author response: thank you for this valuable suggestion. We discussed other countries’ healthcare inequity as follows:

“…Healthcare inequities among the migrant population have also been described in other countries in Latin America. For instance, in Peru, a high proportion of Venezuelan migrants with self-reported chronic diseases were not receiving treatment, and among those who did, 11.5% did not receive it frequently enough. Moreover, among all migrants seeking medical care, more than half preferred pharmacy attention, self-medication, and primary health care. However, a low proportion reported having experienced discrimination in health facilities (44). Another study exploring health care access among migrants in transit from Honduras, El Salvador, and Guatemala revealed they seldom used the public health system. When medical attention was needed, 25.9% could not receive timely attention and those who did report going to an informal health care facility or a private hospital. Diverse barriers were identified, as migrants often reported a lack of information since they usually were unaware of where to ask for attention or the facility location, were avoiding the police, did not have financial resources, or were discriminated against because of their migratory status (45). Another source of inequities is the implementation gap of health-related policies and access to the health care system in countries such as Colombia and Mexico, which recently started receiving growing intra-regional immigration fluxes, and migratory status ends up conditioning access (46). Similarly, the bureaucratic process, high cost, and poor intersectoral coordination in Costa Rica have led to regularization difficulties for Nicaraguan migrants, thus impeding them from obtaining health insurance. Additionally, when insured, they report facing exclusion and xenophobia (47). Similar issues have been identified by medical providers in Uruguay, reporting poor knowledge of the regulatory framework, administrative barriers, differential treatment, and lack of intercultural competence when afro-Caribbean migrant women need medical attention (48). Lastly, during the COVID-19 pandemic, inequities have been increasing in Latin America. Migrants have faced difficulties in obtaining timely information, lack of tailored interventions, continuity of care, and barriers to preventive measures (49). Therefore, the design and implementation of evidence-based health policies with a regional perspective are needed in response to the current inequities observed in different countries in the region…”
